# Kin cell lysis is a danger signal that activates antibacterial pathways of *Pseudomonas aeruginosa*

Michele LeRoux[1,2], Robin L Kirkpatrick[1], Elena I Montauti[1], Bao Q Tran[3], S Brook Peterson[1], Brittany N Harding[1], John C Whitney[1], Alistair B Russell[1], Beth Traxler[1], Young Ah Goo[3], David R Goodlett[3], Paul A Wiggins[4,5], Joseph D Mougous[1]*

[1]Department of Microbiology, University of Washington, Seattle, United States; [2]Molecular and Cellular Biology Program, University of Washington, Seattle, United States; [3]Department of Pharmaceutical Sciences, School of Pharmacy, University of Maryland, Baltimore, United States; [4]Department of Physics, University of Washington, Seattle, United States; [5]Department of Bioengineering, University of Washington, Seattle, United States

**Abstract** The perception and response to cellular death is an important aspect of multicellular eukaryotic life. For example, damage-associated molecular patterns activate an inflammatory cascade that leads to removal of cellular debris and promotion of healing. We demonstrate that lysis of *Pseudomonas aeruginosa* cells triggers a program in the remaining population that confers fitness in interspecies co-culture. We find that this program, termed *P. aeruginosa* response to antagonism (PARA), involves rapid deployment of antibacterial factors and is mediated by the Gac/Rsm global regulatory pathway. Type VI secretion, and, unexpectedly, conjugative type IV secretion within competing bacteria, induce *P. aeruginosa* lysis and activate PARA, thus providing a mechanism for the enhanced capacity of *P. aeruginosa* to target bacteria that elaborate these factors. Our finding that bacteria sense damaged kin and respond via a widely distributed pathway to mount a complex response raises the possibility that danger sensing is an evolutionarily conserved process.

*For correspondence:
mougous@uw.edu

## Introduction

Bacteria can occupy highly dynamic environments, where survival is linked to the ability to sense and respond to an assortment of threats (*Cornforth and Foster, 2013*). It is increasingly clear that in addition to well-understood environmental and nutritive stresses, antagonistic factors elaborated by other bacteria are a common threat that bacteria must cope with (*Little et al., 2008*; *Hibbing et al., 2010*). A number of antagonistic strategies have been identified, including the production of diffusible factors such as small molecule antibiotics. It has been suggested that sub-inhibitory concentrations of these molecules induce specific changes in bacteria, including the production and distribution of resistance mechanisms (*Hoffman et al., 2005*; *Linares et al., 2006*; *Andersson and Hughes, 2014*). Other antagonistic pathways, such as contact-dependent inhibition (CDI) and the type VI secretion system (T6SS), require cell contact (*Hayes et al., 2010*; *Konovalova and Sogaard-Andersen, 2011*). Kin cells are protected from the toxic proteins delivered by these pathways by specific cognate immunity proteins; however, escape from or defense against these pathways by non-kin is not well understood (*Hood et al., 2010*; *Russell et al., 2011*).

The T6SS is a versatile export machinery that can deliver a wide range of proteinaceous effector molecules from donor to recipient Gram-negative bacterial cells (*Hood et al., 2010*; *Coulthurst,*

**eLife digest** Bacteria live in diverse and changing environments where resources such as nutrients and space are often limited. They have thus evolved many survival strategies, including competitive and cooperative behaviors. In the first case, bacteria antagonize or prevent the growth of other microorganisms competing with them for resources, such as by generating antibiotics that specifically target rivals. During cooperation, bacteria may coordinate the production of compounds that have a shared benefit for members of their community.

In multicellular organisms, some cell types sense harmful microorganisms by the injury they cause in neighboring cells. This triggers a process that can lead to the production of molecules that kill the invaders and factors that promote the repair of cellular damage. An equivalent process has so far not been described for single-celled organisms such as bacteria. However, bacteria often live in structured groups containing many different species. In this type of growth environment, the ability of bacteria to sense when others of their species are attacked and to respond by taking measures to defend themselves could improve their chances of survival.

Now, LeRoux et al. reveal that the bacterium *Pseudomonas aeruginosa* is able to detect 'danger signals' released when neighboring *P. aeruginosa* cells are killed by other bacteria. These signals trigger a response in surviving cells by turning on a pathway that controls a number of antibacterial factors. These include the production of the so-called 'type VI secretion system', a molecular machine that delivers a potent cocktail of antibacterial toxins directly into nearby bacteria.

This process, which LeRoux et al. have named '*P. aeruginosa* response to antagonism', or PARA for short, enables *P. aeruginosa* to thrive when grown with competing bacterial species. *P. aeruginosa* is notorious for infecting the lungs of people with the genetic disease cystic fibrosis, as well as chronic wounds often found in people with diabetes. In both cases, when *P. aeruginosa* is present, the numbers of other, often less harmful organisms, tend to decrease. PARA may be one reason for the success of *P. aeruginosa* in these multi-species infections.

*2013*; *Russell et al., 2014a*). One of the best characterized bacterial targeting T6SSs is the Hcp Secretion Island I-encoded T6SS (H1-T6SS) of *Pseudomonas aeruginosa* (*Hood et al., 2010*). The H1-T6SS transports a cargo of at least seven effectors, termed type VI secretion exported 1–7 (Tse1–7) (*Hachani et al., 2014*; *Whitney et al., 2014*). The outcome of intoxication by these proteins can be lysis or cessation of growth (*LeRoux et al., 2012*; *Li et al., 2012*). Like other species with interbacterial T6SSs, *P. aeruginosa* has the capacity to target cells of its own genotype with the H1 pathway. To inhibit self-intoxication, cognate type VI secretion immunity proteins (Tsi) are produced and localized to the cellular compartment that contains the target of the corresponding effector (*Russell et al., 2013*; *Benz and Meinhart, 2014*; *Durand et al., 2014*; *Russell et al., 2014a*). T6SSs are found in at least three genetically distinct configurations present among multiple bacterial phyla, making it one of the most widespread pathways mediating interbacterial antagonism known (*Russell et al., 2014b*).

Expression and activity of the H1-T6SS is tightly regulated at multiple levels (*Silverman et al., 2012*). Stringent post-transcriptional regulation of the H1-T6SS is achieved through the global activation of antibiotic and cyanide synthesis/regulator of secondary metabolism (Gac/Rsm) pathway (*Goodman et al., 2004*; *Mougous et al., 2006b*). This global regulatory system impacts protein production via RsmA, a CsrA-type protein that binds to target mRNA molecules and generally acts to repress their translation (*Lapouge et al., 2008*). RsmA is modulated by levels of the small RNA (sRNA) molecules *rsmY* and *rsmZ*, which bind to and sequester it from its targets. Transcription of the sRNAs is promoted by phosphorylated GacA, the cognate response regulator of the sensor kinase GacS. Finally, two hybrid sensor kinases, RetS and LadS, acting through GacS, repress or stimulate GacA phosphorylation, respectively (*Ventre et al., 2006*; *Goodman et al., 2009*). Consistent with regulation of T6S by the Gac/Rsm pathway, many of its targets in *P. aeruginosa* and related γ-proteobacteria are involved in the production of social or antagonist factors (*Lapouge et al., 2008*). This theme of Gac/Rsm-dependent modulation of antibiotic activity is exemplified by the defect of *P. fluorescens gac* mutants in bacterial and fungal growth inhibition on plants (*Laville et al., 1992*). Though the precise cues that activate the Gac/Rsm pathway are unknown, Haas et al. have found that

one or more signals accumulate in spent bacterial culture supernatants deriving from both self and non-self organisms (*Dubuis and Haas, 2007*).

Posttranslational regulation by the threonine phosphorylation pathway (TPP) constitutes a second level of control over H1-T6SS activity. In this pathway, phosphorylation of a fork head-associated domain-containing protein, Fha1, triggers apparatus assembly and effector secretion (*Mougous et al., 2007*). PppA, a phosphatase, opposes the activity of PpkA on Fha1, returning the system to the inactive state. Additional components of the TPP, encoded by type VI secretion associated genes Q-T (*tagQ-T*), act upstream of these proteins and are thought to be involved in signal transduction (*Hsu et al., 2009*; *Casabona et al., 2013*). Two signals of the TPP have been proposed: surface-associated growth and membrane perturbation (*Silverman et al., 2011*; *Basler et al., 2013*). The latter signal is thought to underlie the observation that organisms with active T6S or type IV secretion (T4S) are more efficiently targeted by the H1-T6SS than those without (*Ho et al., 2013*). It was proposed that the activity of these apparatuses induces local membrane perturbations in *P. aeruginosa* that are sensed by the TPP, leading to posttranslational activation of the H1-T6SS and enhanced recipient cell death (*Basler et al., 2013*). A caveat of these studies is that the *P. aeruginosa* strain used bears an inactivating mutation in *retS*, which constitutively activates the Gac/Rsm pathway, potentially masking the contribution of this major regulatory mechanism to the defense mounted by *P. aeruginosa* against the antagonistic pathways of competing bacteria.

Here, we show that lysed kin cells act as a danger signal that is sensed by the Gac/Rsm pathway of wild-type *P. aeruginosa*. Our experiments provide a mechanism for T6S-dependent killing of competitor bacteria possessing either the T6 or T4 secretion pathways, as both induce *P. aeruginosa* lysis, stimulate the Gac/Rsm pathway, and lead to posttranscriptional de-repression of the H1-T6SS. These findings provide a rationale for the regulation of promiscuous antibiotic mechanisms by a pathway that can respond to self-derived signals.

## Results

### *P. aeruginosa* T6 activity is stimulated by the presence of a non-self organism

Prior work suggests that efficient T6S-dependent effector delivery between *P. aeruginosa* cells (self-targeting) requires deletion of *retS*, whereas this activating mutation is not required for robust T6S-dependent intoxication of other bacteria by *P. aeruginosa* (non-self targeting) (*Hood et al., 2010*; *Russell et al., 2011*; *Ho et al., 2013*). To quantify these observations, we performed bacterial growth competition experiments with wild-type (PAO1) and Δ*retS* with both self- and non-self recipients under comparable conditions. *Burkholderia thailandensis* (*B. thai*) was used as the non-self competitor for these studies, while a *P. aeruginosa* background lacking four H1-T6SS effector–immunity pairs (Δ*tse1-4* Δ*tsi1-4*) was employed as the self recipient. T6S-dependent fitness was determined by comparing the competitive index of a wild-type donor to that of a donor strain lacking *tssM*, which encodes a core structural component of the T6S apparatus (*Felisberto-Rodrigues et al., 2011*). We found that both *P. aeruginosa* wild-type and Δ*retS* strains reduced populations of *B. thai* in a T6S-dependent manner by several orders of magnitude; however, only *P. aeruginosa* Δ*retS* displayed T6S-dependent fitness in co-culture with self recipients (*Figure 1*). This direct comparison of H1-T6SS-dependent antibiosis by *P. aeruginosa* wild-type and Δ*retS* demonstrates that unlike self-intoxication, non-self targeting by the pathway does not require activation of the system by relief of negative regulation. A parsimonious explanation for these data is that the presence of a non-self competitor stimulates H1-T6SS activity.

### Expression and activity of the H1-T6SS is enhanced by T6S of competitor organisms

To test the hypothesis that H1-T6SS activity is affected by the presence of a non-self organism, we used time-lapse fluorescence microscopy (TLFM) in conjunction with customized cell and protein tracking software to monitor the expression and subcellular localization of the conserved T6S ATPase ClpV1 (*Cascales and Cambillau, 2012*; *LeRoux et al., 2012*). Previous studies have established that translational fusion of *gfp* to the 3′ end of *P. aeruginosa clpV1*, encoded within the H1-T6SS gene cluster, yields a stable and functional chimera (*Mougous et al., 2006a*). The fluorescence intensity of ClpV1–GFP provides readout of H1-T6SS expression and the assembly of ClpV1–GFP into punctate

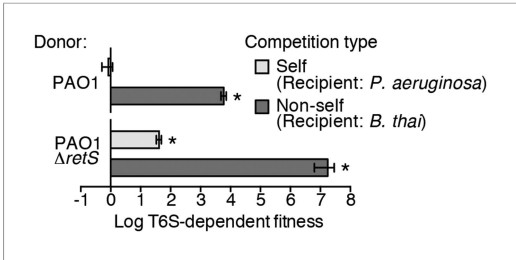

**Figure 1**. Wild-type *P. aeruginosa* cells display a strong T6S-dependent fitness advantage in co-culture with non-self but not self competitors. Outcome of growth competition experiments measuring fitness of *P. aeruginosa* PAO1 parental or Δ*retS* strains in co-cultures with self or non-self recipients under T6SS-promoting conditions. The self recipient was *P. aeruginosa* Δ*tse1-4* Δ*tsi1-4* in the strain background corresponding to the donor genotype (PAO1 or PAO1 Δ*retS*). T6S-dependent fitness was parental donor competitive index (change [final/initial] in ratio of donor and recipient colony forming units [c.f.u.]) normalized to Δ*tssM1* competitive index. Error bars represent ±standard deviation (SD); n = 3 co-cultures. Asterisks denote a fitness advantage significantly >1 (p < 0.01).

foci correlates with activity of the system (*Mougous et al., 2007*; *Kapitein et al., 2013*). Surprisingly, we observed significantly elevated ClpV1–GFP levels in *P. aeruginosa*–*B. thai* co-cultures relative to *P. aeruginosa* monocultures, suggesting that *B. thai* stimulates H1-T6SS expression (*Figure 2A,B*, *Figure 2—figure supplement 1*, *Video 1*). We obtained similar results from a strain bearing a chromosomal, functional translational fusion of *gfp* to *fha1* (*fha1-gfp*), a locus found on the second major H1-T6SS transcript (*Figure 2A,C*, *Figure 2—figure supplement 1*, *Video 2*) (*Mougous et al., 2006a*, *2007*). To determine whether elevated H1-T6SS arises from increased expression throughout the population or from high expression within a subset of *P. aeruginosa*, we examined ClpV1-GFP at the level of individual cells. Consistent with a population-wide, graded response, we found that ClpV1-GFP exhibits a normal distribution both in monoculture and when *P. aeruginosa* is co-cultivated with *B. thai* (*Figure 2—figure supplement 2*). Mirroring the observed trends in expression, after an initial increase in foci frequency—a correlate of T6S activity—associated with growth on a surface (*Silverman et al., 2011*), a larger percentage of *P. aeruginosa* cells grown in the presence of *B. thai* contained ClpV1–GFP and Fha1–GFP foci compared to those grown in monoculture (*Figure 2D,E*). To investigate the generality of these effects on the H1-T6SS, we repeated our experiments using the γ-proteobacterium *Enterobacter cloacae* as the competing organism. *P. aeruginosa* co-cultivated with *E. cloacae* also displayed increased ClpV1–GFP levels and higher foci frequency in comparison to *P. aeruginosa* in monoculture, indicating that this response is not specific to *B. thai* (*Figure 2—figure supplement 3*; *Video 3*).

Previous studies demonstrated that the H1-T6SS of *P. aeruginosa* targets recipient cells that possess an active T6SS with greater efficiency than those that do not (*LeRoux et al., 2012*; *Basler et al., 2013*). Though this behavior was shown for *P. aeruginosa* Δ*retS*, and not wild-type cells, the apparent capacity of the organism to sense the T6SS of a recipient cell prompted us to test whether T6S in non-self competitors is involved in stimulation of the H1 pathway. Indeed, we found that *B. thai* and *E. cloacae* strains bearing in-frame deletions of the *tssM* genes associated with their known antibacterial T6S pathways, are unable to stimulate T6S in *P. aeruginosa* (*Figure 2*, *Figure 2—figure supplement 1*, *Figure 2—figure supplement 3*, *Videos 1–3*) (*Schwarz et al., 2010*; *Koskiniemi et al., 2013*; *Whitney et al., 2014*). Changes in *P. aeruginosa* growth rate dependent upon recipient T6S could influence gene expression and protein accumulation, potentially accounting for altered expression and activity of H1-T6SS proteins. However, this possibility was excluded by our observation that the doubling time of *P. aeruginosa* is insensitive to the activity of the bacterial cell-targeting T6S of *B. thai* (T6S[BT]) (*Figure 2—figure supplement 4*). Taken together, these data demonstrate that co-cultivation of *P. aeruginosa* with a non-self organism possessing an active T6S apparatus leads to elevated expression and activity of the H1-T6SS. Henceforth, we refer to this response of *P. aeruginosa* as PARA (*P. aeruginosa response to antagonism*).

## PARA does not require the TPP

A recent report implicated the TPP in the capacity of *P. aeruginosa* to sense and respond to T6S in target cells (*Basler et al., 2013*). To determine if PARA requires the TPP, we monitored the T6S-dependent response of *P. aeruginosa* bearing a deletion in *pppA*, which encodes a TPP-associated phosphatase (*Mougous et al., 2007*). Interestingly, though this deletion was previously shown to

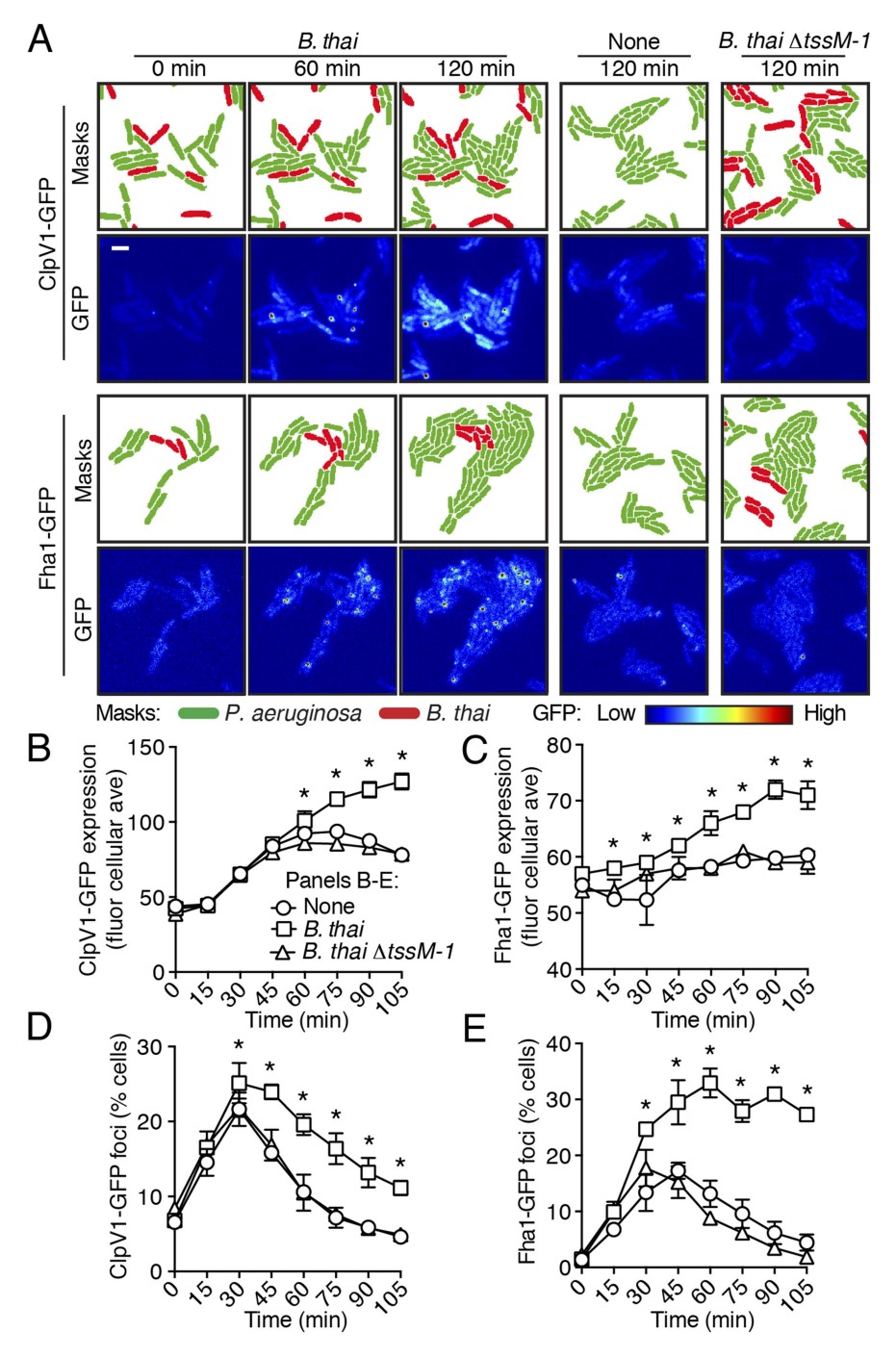

**Figure 2**. Non-self competitor bacteria stimulate expression and activity of the H1-T6SS. (**A**) H1-T6SS expression is increased in *P. aeruginosa* co-cultured with *B. thai* expressing an active T6SS. Time-lapse fluorescence microscopy (TLFM) sequences of *P. aeruginosa clpV1-gfp* (upper) or *fha1-gfp* (lower) in monoculture or in co-culture with the indicated competitor. Cropped regions from representative time-points are displayed. Remaining time points for monoculture and co-culture with *B. thai ΔtssM-1* are depicted in *Figure 2—figure supplement 1*; see also *Videos 1 and 2*. Masks colored by cell identity depict automated cell identification generated from the phase image. Scale bar, 6 μm. (**B–C**) Quantification of H1-T6SS expression from the *P. aeruginosa clpV1-gfp* (**B**) and *fha1-gfp* (**C**) monoculture and co-culture TLFM experiments described in (**A**). Average cellular GFP intensity for *P. aeruginosa* cells was calculated from background-subtracted images. (**D–E**) H1-T6SS activity is increased in the presence of *B. thai* with an active T6SS. Percentage of *P. aeruginosa clpV1-gfp* (**D**) or *fha1-gfp* (**E**) cells with GFP foci for experiments
*Figure 2. continued on next page*

*Figure 2. Continued*

described in (**A**). Error bars represent ±SD; n = 3 fields. Asterisks indicate significant differences when *B. thai* was present (p < 0.05).
The following figure supplements are available for figure 2:

**Figure supplement 1**. Competitors require an active T6SS to stimulate the *P. aeruginosa* H1-T6SS.

**Figure supplement 2**. Increased H1-T6SS expression occurs throughout the population.

**Figure supplement 3**. *E. cloacae* stimulates the H1-T6SS of *P. aeruginosa* in a T6S-dependent manner.

**Figure supplement 4**. *P. aeruginosa* doubling time is not affected by the presence of *B. thai*.

abrogate the ability of *P. aeruginosa* Δ*retS* to respond differentially to bacteria containing or lacking a functional T6SS, we found that in the wild-type background, Δ*pppA* retains the capacity to respond to T6S[BT] (*Figure 3A*, *Figure 3—figure supplement 1A*). PppA is a negative regulator of the TPP; therefore, we sought to rule out the possibility that the remaining TPP components, TagQ-T and PpkA, are sufficient to mediate PARA. Inactivation of all known components of the TPP results in a failure to assemble an active T6S apparatus, which would confound our analyses. However, activity can be restored to TPP deficient strains by the deletion of a gene encoding an independent post-translational repressor of the H1-T6SS, TagF (*Silverman et al., 2011*). As observed with Δ*pppA*, the H1-T6SS of a *P. aeruginosa* strain lacking *tagF* and all known TPP components (ΔTPP Δ*tagF*) was activated by *B. thai* in a T6S[BT]-dependent manner (*Figure 3B*, *Figure 3—figure supplement 1B*). Consistent with measurements of expression and activation, growth competition experiments revealed that wild-type, Δ*pppA*, and Δ*tagF* ΔTPP strains of *P. aeruginosa* intoxicate *B. thai* and *E. cloacae* with active interbacterial T6SSs more efficiently than those without (*Figure 3C*, *Figure 3—figure supplement 2*). These data indicate that the TPP is not required for PARA.

It is worth noting that several lines of evidence suggest PARA is a process distinct from the previously characterized intercellular T6-based response referred to as 'T6SS dueling' (*Basler and Mekalanos, 2012*). First, PARA is accompanied by changes to T6S gene expression, whereas dueling is a posttranslational behavior thought to involve rapid changes in protein localization. Second, the TPP was found to be essential for T6SS dueling (*Basler et al., 2013*); however, this pathway is dispensable for PARA. Finally, spatiotemporal coordination of ClpV1-GFP-containing foci forms the basis for T6SS dueling, while such paired focus events are not correlated with PARA (data not shown). Motivated by the finding of PARA as a distinct pathway of clear functional relevance to interbacterial interactions, we proceeded to investigate its mechanistic underpinnings.

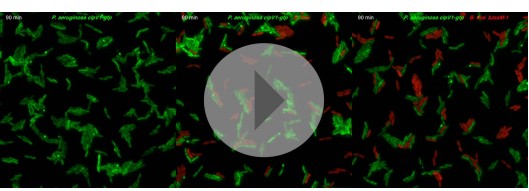

**Video 1.** ClpV1 expression increases in the presence of *B. thai* bearing an active T6SS. TLFM sequences of ClpV1-GFP in *P. aeruginosa* during monoculture or interspecies co-culture. *P. aeruginosa clpV1-gfp* cells without competitor (left sequence), with *B. thai mCherry* (middle sequence), or with *B. thai* Δ*tssM-1 mCherry* (right sequence) were imaged at 15 min intervals. Overlays of GFP and mCherry channels are displayed. The same thresholds were applied to all background-subtracted GFP channels. See *Figure 2B,D* for quantification.

## PARA is a multifaceted response that requires the Gac/Rsm pathway

The H1-T6SS is regulated at the transcriptional level by the quorum sensing regulator LasR, and at the posttranscriptional level by RsmA and RsmF, RNA binding proteins that directly mediate the effects of Gac/Rsm signaling (*Figure 4A*) (*Brencic and Lory, 2009*; *Lesic et al., 2009*; *Marden et al., 2013*). As a first step toward defining the pathway through which PARA operates, we measured H1-T6SS transcription and translation in response to co-culture with

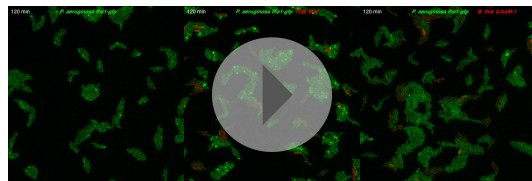

**Video 2.** Fha1 expression increases in the presence of *B. thai* bearing an active T6SS. TLFM sequences depicting expression of Fha1-GFP in *P. aeruginosa* during monoculture or interspecies co-cultures. *P. aeruginosa fha1-gfp* cells without competitor (left sequence), with *B. thai* mCherry (middle sequence), or with *B. thai* ∆*tssM-1* mCherry (right sequence) were imaged at 15 min intervals. Overlays of GFP and mCherry channels are displayed. The same thresholds were applied to all background-subtracted GFP channels. Quantification is provided in **Figure 2C,E**.

*B. thai* using previously characterized chromosomal β-galactosidase reporters (**Brencic and Lory, 2009**). In contrast to minor effects on transcription, we found that co-culture of *P. aeruginosa* with *B. thai* markedly stimulates H1-T6SS translation (**Figure 4B**). In control experiments with *B. thai* ∆*tssM-1* as the competitor, we observed no enhancement of H1-T6SS expression. Based on these data, we hypothesized that PARA is driven by the Gac/Rsm pathway.

If the Gac/Rsm pathway was involved in PARA-mediated H1-T6SS activation, we would expect transcription of its associated sRNA molecules to be elevated in cells co-cultivated with *B. thai* (**Figure 4A**). Indeed, using a chromosomal fluorescent reporter of *rsmZ* transcription, we found *B. thai* stimulates expression of this sRNA on a time scale consistent with other effects associated with PARA (**Figure 4C,D**, **Figure 4—figure supplement 1**, **Video 4**). Cells exposed to *B. thai* ∆*tssM-1* did not exhibit changes in *rsmZ* expression. To further explore the link between Gac/Rsm and PARA, we measured the expression of a validated direct target of RsmA that is unrelated to the H1-T6SS, *magA* (**Brencic and Lory, 2009**; **Robert-Genthon et al., 2013**). As observed for H1-T6SS reporters, a translational chromosomal fusion of the *magA* promoter to *gfp* (p-*magA-gfp*) displayed increased expression in response specifically to co-cultivation with *B. thai* bearing an active interbacterial T6SS (**Figure 4E**). Taken together with our β-galactosidase reporter results, these data show that PARA is a Gac/Rsm-mediated posttranscriptional response of *P. aeruginosa* to T6S in other bacteria.

Next we examined upstream components of the Gac/Rsm pathway that might mediate PARA. In *P. aeruginosa*, the output of the Gac/Rsm pathway is modulated by three sensor kinases (**Figure 4A**) (**Jimenez et al., 2012**). We began by investigating GacS, which directly phosphorylates GacA, the response regulator that activates *rsmY* and *rsmZ* expression (**Figure 4A**). Consistent with our hypothesis that PARA requires Gac/Rsm mediated signaling, we found that a strain bearing an in-frame deletion of *gacS* fails to elevate H1-T6SS expression in response to T6S^BT (**Figure 4F**). The remaining Gac/Rsm-associated sensor kinases, RetS and LadS, play accessory roles in the Gac/Rsm pathway by positively or negatively regulating GacS phosphorylation of GacA, respectively (**Figure 4A**). Our experiments showed that strains lacking *ladS* respond to T6S^BT similarly to the wild-type (**Figure 4G**, **Figure 4—figure supplement 2A**); however, we found that PARA is abrogated in strains lacking *retS* (**Figure 4H**, **Figure 4—figure supplement 2B**). This finding suggests that the relief of RetS repression of GacS-catalyzed phosphorylation of GacA is important in PARA induction. This might occur through direct binding of RetS to a signal produced by the presence of the competitor organism, or, alternatively, RetS could transduce a signal from an upstream sensor. To investigate these possibilities, we generated a *P. aeruginosa* strain bearing a chromosomally-encoded RetS

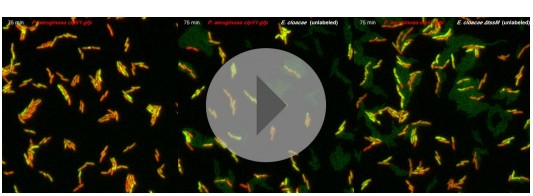

**Video 3.** ClpV1 expression increases in the presence of *E. cloacae* bearing an active T6SS. TLFM sequences depicting expression of ClpV1-GFP in *P. aeruginosa* during monoculture or interspecies co-cultures. *P. aeruginosa clpV1-gfp* mCherry cells without competitor (left sequence), with *E. cloacae* (middle sequence), or with *E. cloacae* ∆*tssM* (right sequence) were imaged at 15 min intervals. Overlays of GFP and mCherry channels are displayed. *P. aeruginosa* cells were labeled with constitutive mCherry and therefore the overlay of GFP and mCherry appears yellow. The unlabeled *E. cloacae* cells are visible (light green) due to autofluorescence in the GFP channel. The same thresholds were applied to all background-subtracted GFP channels. Quantification is provided in **Figure 2—figure supplement 3**.

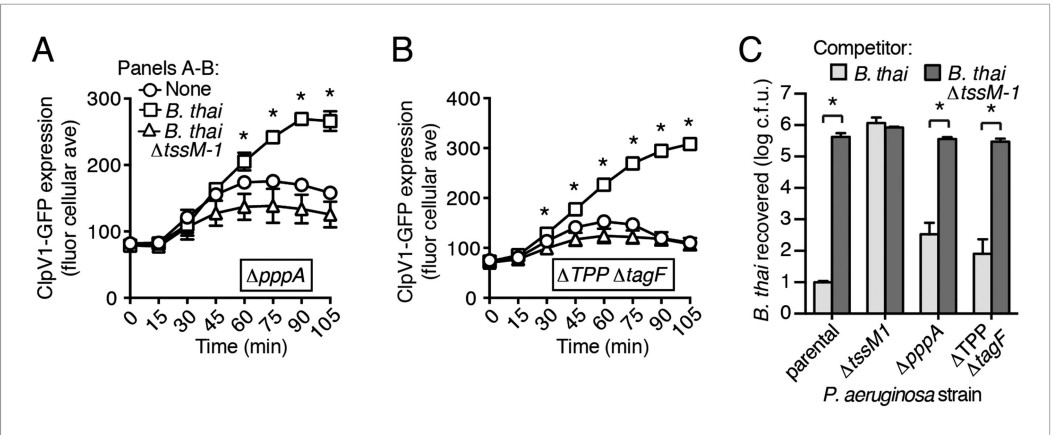

**Figure 3**. PARA does not require the TPP. (**A–B**) Increased H1-T6SS expression in the presence of *B. thai* does not require a functional TPP. Average ClpV1-GFP cellular fluorescence intensity in *P. aeruginosa clpV1-gfp ΔpppA* (**A**) and *ΔTPP ΔtagF* (**B**) backgrounds during monoculture or co-culture with the indicated competitors. Error bars represent ±SD; n = 3 fields. Asterisks indicate significant differences when *B. thai* was present (p < 0.05). Corresponding H1-T6SS activity is shown in *Figure 3—figure supplement 1*. (**C**) The TPP is not required for preferential targeting of *B. thai* with an active T6SS. Outcome of growth competition experiments measuring survival of *B. thai* following co-culture with the indicated *P. aeruginosa* strain under T6SS-promoting conditions. Error bars represent ±SD; n = 3 co-cultures.
The following figure supplements are available for figure 3:

**Figure supplement 1**. Elevated H1-T6SS activity in the presence of *B. thai* does not require the TPP.

**Figure supplement 2**. *P. aeruginosa* does not require the TPP to differentially target *E. cloacae* with a T6SS.

variant containing an amino acid substitution of a highly conserved residue in the predicted periplasmic signal binding pocket (W90A) (*Jing et al., 2010*; *Vincent et al., 2010*). Consistent with RetS acting as the direct sensor for PARA, in the *retS*[W90A] background, neither ClpV1 expression levels nor activity were affected by co-cultivation with *B. thai* (*Figure 4I*, *Figure 4—figure supplement 2C*). A general disruption of RetS function cannot be excluded; however, the finding that H1-T6SS expression levels in *retS*[W90A] do not approach those detected in a *ΔretS* mutant demonstrates that this allele retains partial function. Altogether, our findings suggest that RetS functions upstream in the Gac/Rsm pathway to mediate PARA.

Gac/Rsm is a pathway generally noted as a regulator of antibiosis; its stimulation in pseudomonads can increase the expression of a variety of antibiotic factors in addition to T6S, including hydrogen cyanide, secreted hydrolytic enzymes, and phenazines (*Lapouge et al., 2008*). We reasoned that the fitness of cells undergoing PARA is derived not only from an increase in expression and activity of the H1-T6SS, but also from increased levels of these co-regulated factors. To test this, we used growth competition assays and TLFM co-cultures to compare the fitness of *ΔgacS* to a strain lacking only the function of the H1-T6S pathway (*ΔtssM1*). Consistent with our hypothesis, the fitness of *ΔgacS* was reduced beyond that of *ΔtssM1* only when in competition with either *B. thai* (34-fold reduction) or *E. cloacae* (15-fold reduction) bearing an active interbacterial T6S pathway (*Figure 5A,B*). TLFM experiments further indicated a substantial increase in lysis of *P. aeruginosa ΔgacS* relative to wild-type or *ΔtssM1* (*Figure 5C*, *Video 5*). These phenotypes are not due to a general growth defect, as neither *P. aeruginosa ΔgacS* nor *ΔtssM1* exhibit a competitive defect when grown in co-culture with the parental strain (*Figure 5D*). In total, these data indicate that PARA is a complex bacterial defense mechanism comprising the H1-T6SS and other Gac/Rsm-regulated factors of *P. aeruginosa*.

## T6S-dependent interactions result in release of a diffusible signal

T6S-dependent interactions require direct cell–cell contact; therefore, we asked whether PARA is also contact-dependent. To examine this, we computationally sorted *P. aeruginosa* into populations contacting or not contacting *B. thai* during the course of TLFM experiments (*Figure 6A*). Surprisingly,

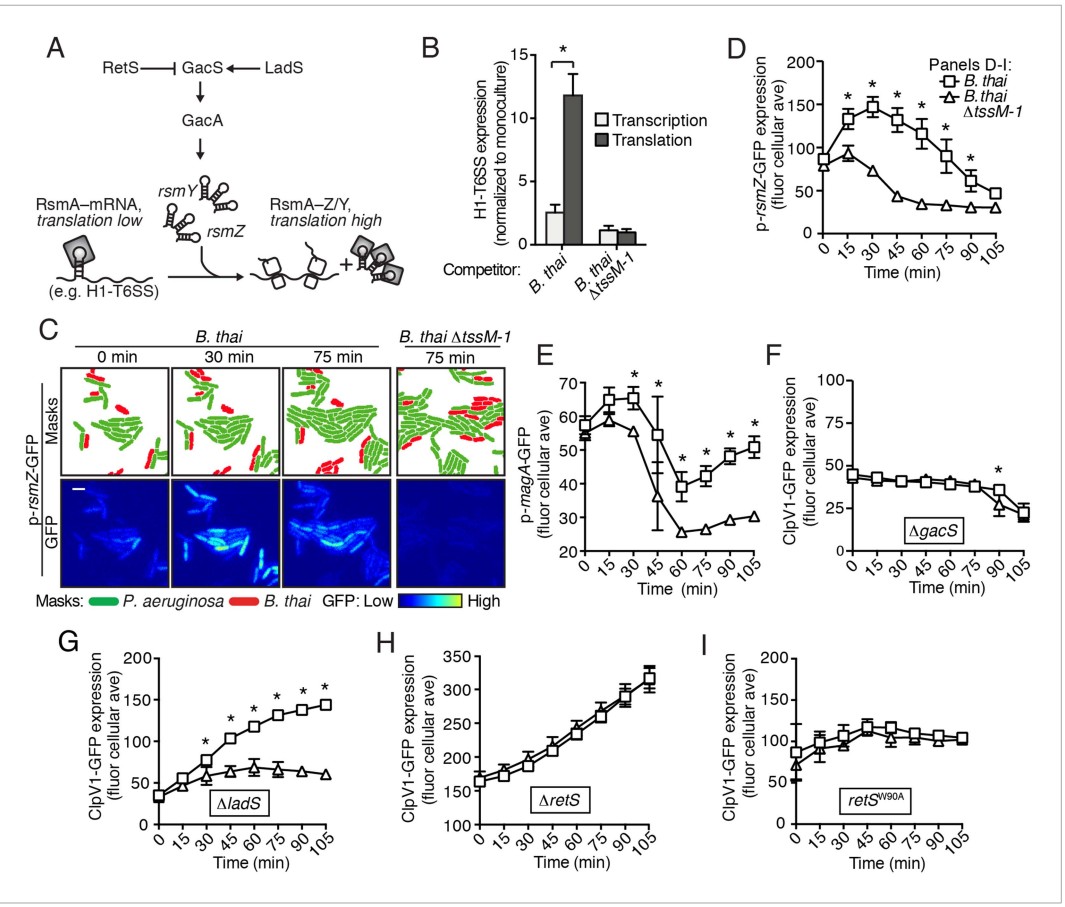

**Figure 4**. The Gac/Rsm pathway is required for PARA. (**A**) Schematic depicting the Gac/Rsm pathway of *P. aeruginosa*. The orphan sensor kinases RetS and LadS exert opposing activity on a third sensor kinase, GacS, which in turn activates its cognate response regulator, GacA. Once active, GacA promotes increased transcription of the small RNAs *rsmY* and *rsmZ*. These molecules bind and sequester RsmA; therefore, when abundant, they prevent RsmA binding and destabilization of target mRNAs, including H1-T6SS transcripts. (**B**) Elevated H1-T6SS expression in the presence of *B. thai* occurs primarily at the post-transcriptional level. *P. aeruginosa* strains bearing chromosomally encoded transcriptional or translational fusions to *tssA1* (**Brencic and Lory, 2009**) were incubated with the indicated competitor. Fold H1-T6SS increase in expression was determined by normalizing *P. aeruginosa* co-cultures to the corresponding strain cultivated in monoculture. n = 3 co-cultures; asterisk indicates significant differences between translational and transcriptional activity (p < 0.05). (**C**) *RsmZ* expression is elevated in the presence of *B. thai* containing a T6SS. Cells masks and the GFP fluorescence channel from representative TLFM sequences of the indicated *P. aeruginosa* p-*rsmZ-gfp* co-cultures. Additional time points are shown in **Figure 4—figure supplement 1**; see also **Video 4**. (**D**) Average cellular fluorescence intensity from *P. aeruginosa* p-*rsmZ-gfp* corresponding to (**C**). (**E**) Expression of MagA is elevated in the presence of T6S^BT. Average cellular GFP intensity of *P. aeruginosa* p-*magA-gfp* in co-culture with *B. thai* or *B. thai* Δ*tssM-1*. (**F**) GacS is required for H1-T6SS activation in response to T6S^BT. ClpV1-GFP expression was quantified for co-cultures of *P. aeruginosa* Δ*gacS clpV1-gfp* with the indicated *B. thai* competitor. (**G**) LadS is not required for elevated H1-T6SS expression in the presence of *B. thai*. Average cellular ClpV1-GFP intensity of *P. aeruginosa* Δ*ladS clpV1-gfp* in co-culture with the indicated competitor. See **Figure 4—figure supplement 2A** for H1-T6SS activity. (**H**) *P. aeruginosa* cells lacking *retS* are unable to respond to T6S^BT. Average cellular ClpV1-GFP intensity of *P. aeruginosa* Δ*retS clpV1-gfp* in co-culture with the indicated competitor. See **Figure 4—figure supplement 2B** for H1-T6SS activity. (**I**) A conserved residue in the periplasmic domain of RetS is required for *P. aeruginosa* response to T6S^BT. Average cellular ClpV1-GFP intensity of *P. aeruginosa clpV1-gfp retS*^W90A cultivated in the presence of the indicated competitor. See **Figure 4—figure supplement 2C** for H1-T6SS activity. n = 3 fields; asterisks indicate significant differences when *B. thai* was present (p < 0.05).

The following figure supplements are available for figure 4:

**Figure supplement 1**. *RsmZ* expression is not stimulated by *B. thai* lacking an active T6SS.

*Figure 4. continued on next page*

*Figure 4. Continued*

**Figure supplement 2**. PARA-associated increases in H1-T6SS activity depend on RetS but not LadS.

we found no significant difference in ClpV1-GFP levels between these populations of cells (*Figure 6B*), suggesting that either PARA does not require immediate cell contact between *P. aeruginosa* and a competitor, or *P. aeruginosa*–competitor contacts can be sensed by non-contacting cells. To differentiate between these possibilities, we employed a population-level approach in which we measured PARA-associated phenotypes in *P. aeruginosa* cells following cultivation on an agar plate (*Figure 6C*). Consistent with our microscopy experiments, PARA was detected in *P. aeruginosa* cultivated in the presence of *B. thai,* but not in monoculture or with *B. thai ΔtssM-1* (*Figure 6D,E*; Condition 1). We next measured PARA in *P. aeruginosa* cells separated by a membrane from either a *B. thai* monoculture or a *P. aeruginosa*–*B. thai* co-culture. Strikingly, PARA was detected when *P. aeruginosa* was adjacent to contacting *P. aeruginosa*–*B. thai* mixtures, but not when adjacent to *B. thai* alone (*Figure 6D,E*; Conditions 2 and 3). Combined with our microscopy results, these data strongly suggest that during co-culture with *P. aeruginosa*, the activity of T6S^BT, or T6S^BT itself, generates a diffusible molecule that triggers PARA in surrounding cells.

## Self-derived lysate is sufficient to trigger PARA

One outcome of T6S-dependent interactions is cell lysis. Thus, we posited that lysed *P. aeruginosa*—a consequence of T6S^BT activity—could be the source of the diffusible factor mediating PARA. In agreement with this hypothesis, when *P. aeruginosa* is cultivated with *B. thai*, but not *B. thai ΔtssM-1*, there is a significant increase in the number of *P. aeruginosa* cells that undergo lysis (*Figure 7A*). Furthermore, concurrent temporal analysis of H1-T6SS expression and cell lysis in TLFM sequences indicated that *P. aeruginosa* lysis precedes elevation of ClpV1-GFP expression and the initiation of *B. thai* cell death (*Figure 7B*, *Video 6*). To determine if lysed *P. aeruginosa* cells are sufficient to induce PARA, we incubated reporter strains with lysate derived from wild-type *P. aeruginosa*. Lysate derived from *P. aeruginosa*, but not from *B. thai*, stimulated the Gac/Rsm pathway and recapitulated downstream PARA-associated phenotypes (*Figure 7C,D*).

Due to its complex nature, lysate has the potential to lead to non-specific changes in cellular physiology that could confound interpretation of a small set of individual reporters. To gain a more comprehensive view of the effects of self-derived lysate on *P. aeruginosa* physiology, we used quantitative mass spectrometry to measure changes in protein abundance at the proteome level. To our knowledge, the global impact of Gac/Rsm activation on the *P. aeruginosa* proteome has not been reported. In lieu of this, we utilized a microarray study published by Lory and colleagues (Δ*retS* vs wild-type) in order to generate a list of proteins under Gac/Rsm control. Given the short time scale of our experiment and the relatively slow rate by which many proteins are recycled, we focused on factors positively regulated by Gac/Rsm. Remarkably, despite constituting only 4.5% of the proteins detected in our proteome, known Gac/Rsm targets accounted for 49% of proteins induced greater than twofold by the addition of lysate (*Figure 7E*). We performed a gene set enrichment analysis and found that both Gac/Rsm regulated proteins and H1-T6SS proteins are significantly enriched in cells treated with lysate (Gac/Rsm: NES, 2.1, FDR ≤ 0.2%, p > 0.01; H1-T6SS: NES, 1.6, FDR ≤ 5.1%, p < 0.01) (*Mootha et al., 2003*; *Subramanian et al., 2005*).

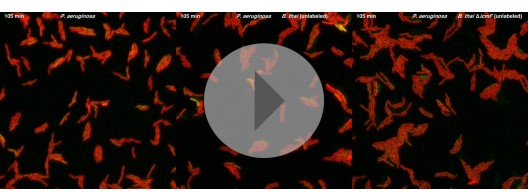

**Video 4.** *RsmZ* expression is elevated in the presence of *B. thai* bearing an active T6SS. TLFM sequences depicting expression of *rsmZ* in *P. aeruginosa* during interspecies co-cultures. *P. aeruginosa* p-*rsmZ-gfp mCherry* cells with *B. thai* (left sequence) and *B. thai ΔtssM* (right sequence) were imaged at 15 min intervals. Overlays of GFP and mCherry channels are displayed. *P. aeruginosa* cells were labeled with constitutive mCherry, thus cells appear yellow. The unlabeled *B. thai* cells are visible (light green) due to autofluorescence in the GFP channel. The same thresholds were applied to all background-subtracted GFP channels. See *Figure 4D* for quantification.

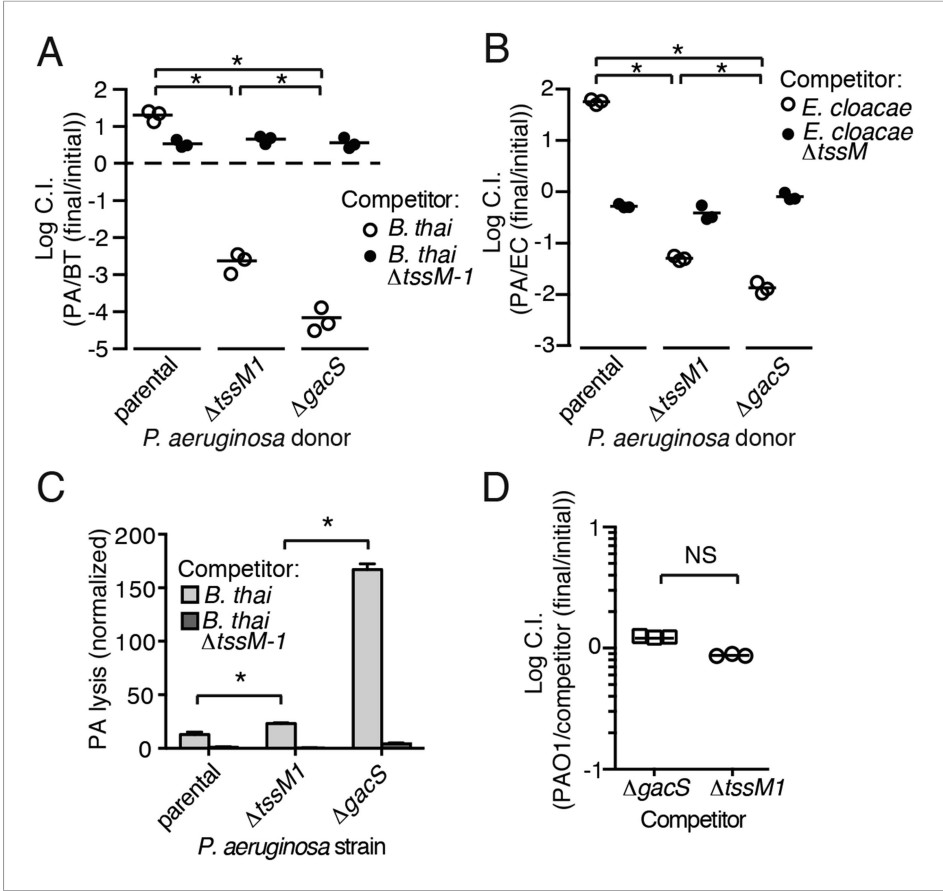

**Figure 5**. Disruption of the Gac/Rsm pathway results in a profound fitness defect in interspecies co-culture. (**A–B**) A *P. aeruginosa* strain with an inactivated Gac/Rsm pathway displays fitness defects beyond a strain lacking H1-T6S. Outcome of interspecies growth competition experiments between the indicated *P. aeruginosa* and *B. thai* (**A**) or *E. cloacae* (**B**) strains. n = 3 co-cultures. C.I., competitive index. PA, *P. aeruginosa*. BT, *B. thai*. EC, *E. cloacae*. (**C**) *P. aeruginosa* lysis promoted by T6S[BT] is increased in a strain lacking a functional Gac/Rsm pathway. *P. aeruginosa* lysis events from TLFM sequences were normalized to initial number of contacts with *B. thai*. See also *Video 5*. n = 3 fields. (**D**) A *gacS* deletion does not alter growth rate. Outcome of intraspecies growth competition experiments between PAO1 and the indicated competitor strains under conditions identical to those used in (**A–B**). n = 3 co-cultures. (**A–D**) Error bars represent ±SD; asterisks indicate significant differences between indicated groups (p < 0.05). NS, not significant.

The remaining 51% of induced proteins could include novel Gac/Rsm targets and lysate-responsive factors outside of the Gac/Rsm regulon. Together, these data show that self-derived lysate activates the Gac/Rsm pathway and is sufficient to induce PARA.

A predicted consequence of PARA is a diminished capacity of *P. aeruginosa* to kill competitor organisms that lack an active lytic pathway. Our finding that lysate is sufficient to induce PARA provided an opportunity to test this directly. Specifically, we asked whether artificial induction of PARA by lysate could stimulate *P. aeruginosa* killing of *B. thai* lacking an active interbacterial T6SS. To this end, we measured the effect of lysate-induced PARA on *P. aeruginosa* fitness in growth competition experiments with *B. thai* Δ*tssM-1*. In agreement with our prediction, the fitness of *P. aeruginosa* increased approximately 2.5-fold in the presence of lysate (*Figure 7F*, *Figure 7—source data 1*). In summary, our data suggest that *P. aeruginosa* cells that undergo lysis as a result of interspecies antagonism serve as a signal for Gac/Rsm-mediated stimulation of antibiosis in the remainder of the population.

## Type IV secretion triggers PARA

Ho et al. recently demonstrated that *E. coli* cells possessing an active IncP-type conjugative type IV secretion (T4S) apparatus are targeted more efficiently by the H1-T6SS of wild-type *P. aeruginosa* than

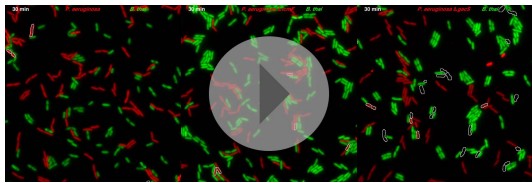

**Video 5.** Inactivation of the Gac/Rsm pathway results in a loss of interbacterial fitness. TLFM sequences of the indicated *P. aeruginosa* mCherry strains cultivated with *B. thai* GFP. Overlays of GFP and mCherry channels are displayed. Lysing *P. aeruginosa* cells are outlined in white. Quantification is provided in *Figure 5C*.

*E. coli* lacking this system (*Ho et al., 2013*). The authors of this study proposed that—like T6SS dueling—this effect is a result of enhanced T6S activity against *E. coli* deriving from local membrane perturbations made by the incoming conjugative apparatus. We hypothesized that the effect observed could also be a result of PARA. To test this, we performed interbacterial growth competition experiments between *P. aeruginosa* and *E. coli*, *E. coli* containing the IncP-type RP4 conjugative plasmid used by Ho et al., or *E. coli* bearing a mutant form of this plasmid lacking a functional T4S apparatus (*ΔtraG*) (*Waters et al., 1992*; *Pansegrau et al., 1994*). Similar to earlier findings, we found that the H1-T6SS of *P. aeruginosa* reduces populations of T4S⁺ *E. coli* to a greater extent than T4S⁻ *E. coli* (*Figure 8A*, *Figure 8—figure supplement 1*). However, contrary to observations made by Ho et al. using the *P. aeruginosa ΔretS* background, we found that removal of the TPP did not abrogate the ability of *P. aeruginosa* to differentially target T4S⁺ and T4S⁻ *E. coli* with the H1-T6SS (*Figure 8A*).

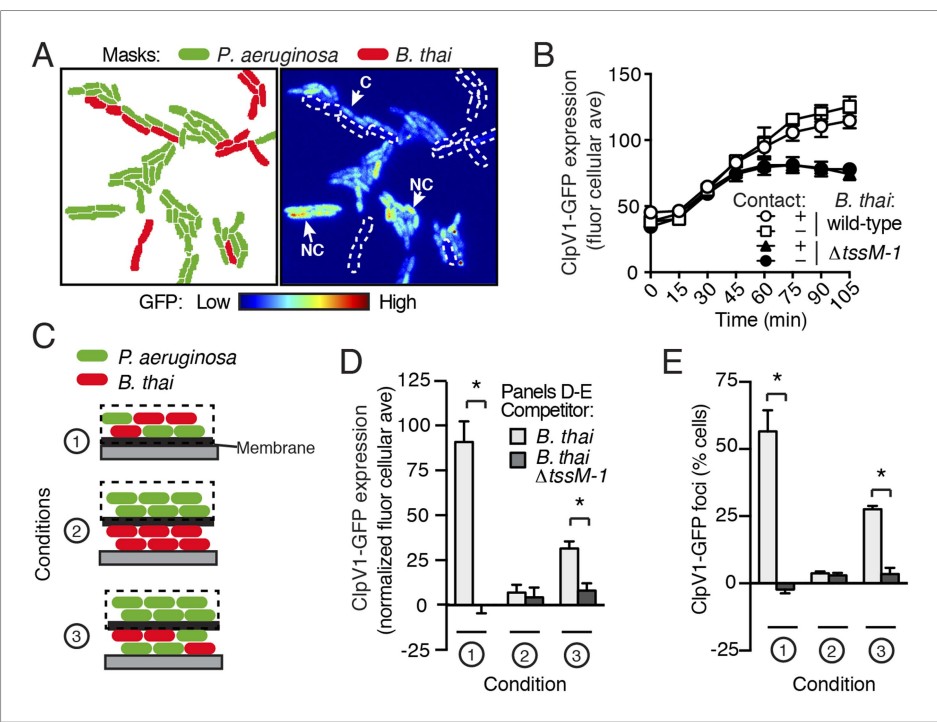

**Figure 6**. PARA is induced by a diffusible signal. (**A**) The PARA-associated increase in H1-T6SS expression is not contact-dependent. A representative region from a *P. aeruginosa clpV1-gfp*–*B. thai* co-culture following 120 min of growth is depicted. Cells masks are colored by cell identity (left panel); GFP intensity with *B. thai* cell positions outlined in white dashed lines (right panel). Arrows indicate *P. aeruginosa* cells contacting (**C**) or not contacting (NC) *B. thai*. (**B**) Average cellular ClpV1-GFP expression for contacting and non-contacting subpopulations described in (**A**). (**C**) Schematic depicting the experimental setup for (**D**). (**D–E**) PARA induction requires proximity to contacting *P. aeruginosa*–*B. thai* cells. Bacterial growth was initiated as pictured in (**C**). ClpV1-GFP was measured in populations on the membrane (black dashed lines). Average cellular ClpV1-GFP expression (**D**) and percentage of cells with foci (**E**) was determined. ClpV1-GFP expression measured in co-cultures was normalized by subtracting *P. aeruginosa* monoculture measurements. Error bars represent ±SD; n = 3 fields. Asterisks indicate significant differences between indicated groups (p < 0.05).

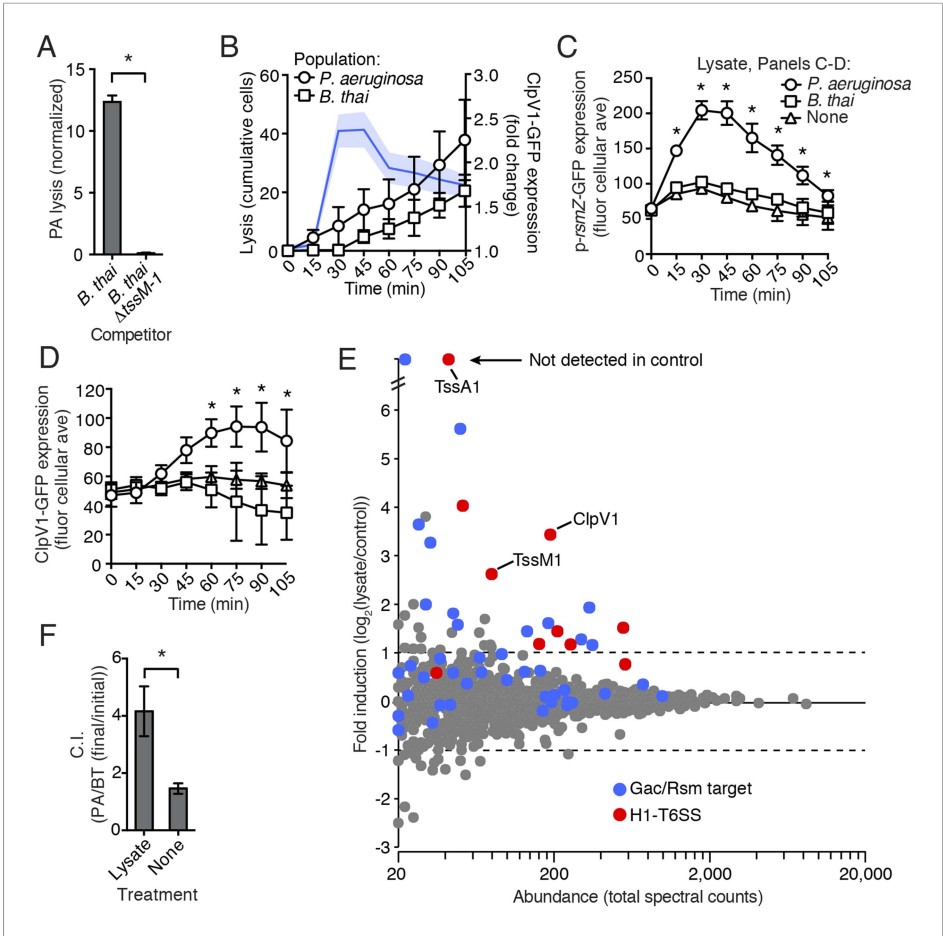

**Figure 7**. *P. aeruginosa* lysis is sufficient to induce PARA. (**A**) T6S$^{BT}$ promotes *P. aeruginosa* lysis. Lysis of *P. aeruginosa* was measured under TLFM conditions and data were normalized to contacts with *B. thai*. n = 3 fields. Asterisk indicates significant difference between *B. thai* and *B. thai* Δ*tssM-1* (p < 0.05). (**B**) *P. aeruginosa* lysis precedes induction of H1-T6SS expression and *B. thai* lysis. Lysis (left axis) and fold increase in ClpV1-GFP (blue line, right axis) measured concurrently under TLFM conditions. ClpV1-GFP levels from *P. aeruginosa*–*B. thai* co-cultures were normalized to *P. aeruginosa* monocultures. Error bars and light blue shading, ± SD. n = 4 fields. See also *Video 6*. (**C**) *P. aeruginosa* lysate stimulates the Gac/Rsm pathway. Average cellular p-*rsmZ*-GFP expression in *P. aeruginosa* cultivated on lysate-infused growth pads. Cellular GFP expression was calculated as described in *Figure 2*. (**D**) H1-T6SS expression is stimulated by *P. aeruginosa* lysate. Average cellular ClpV1-GFP expression of *P. aeruginosa* cultivated on lysate-infused growth pads. (**C–D**) n = 3 fields; asterisks indicated significant differences between *P. aeruginosa* lysate and no lysate (p < 0.05). (**E**) Expression of Gac/Rsm-regulated proteins is increased in lysate-treated *P. aeruginosa* cells. Quantitative mass spectrometry was used to compare the proteome of PBS (control) and lysate treated *P. aeruginosa*. Previously identified Gac/Rsm targets are indicated and H1-T6SS proteins discussed in this study are labeled. Data derive from two biological replicates. (**F**) Lysate stimulates H1-T6SS-mediated killing of *B. thai* Δ*tssM-1*. Outcome of interspecies growth competition experiments between the indicated *P. aeruginosa* and *B. thai* in the presence or absence of *P. aeruginosa*-derived lysate. Error bars represent ±SD; n = 3 co-cultures. Asterisk indicates significant difference between lysate and no lysate treatments (p < 0.05).

The following source data is available for figure 7:

**Source data 1**. Proteins and corresponding spectral counts identified by quantitative mass spectrometry for *P. aeruginosa* cells with and without lysate exposure.

Having ruled out a requirement for the TPP in T4S sensing by wild-type *P. aeruginosa*, we next measured PARA-associated phenotypes in *P. aeruginosa*–*E. coli* co-cultures. These experiments showed that *P. aeruginosa* cultivated with T4S$^+$ *E. coli* exhibits increased *rsmZ* expression and elevated

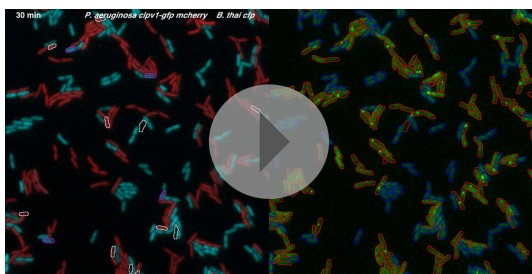

**Video 6.** Lysis of *P. aeruginosa* precedes an increase in H1-T6SS expression and *B. thai* lysis. TLFM sequences depicting lysis (left) and ClpV1-GFP expression (right) in a *P. aeruginosa*–*B. thai* co-culture. A mixture of *P. aeruginosa clpV1-GFP mCherry* and *B. thai* CFP were imaged at 5-min intervals. Right panel displays the background-subtracted GFP channel with *P. aeruginosa* cells outlined in red and *B. thai* outlined in blue. Left panel displays an overlay of mCherry (*P. aeruginosa*) and CFP (*B. thai*) channels; lysing *P. aeruginosa* (white outlines) and lysing *B. thai* (magenta) are indicated. See **Figure 7B** for quantification.

ClpV1–GFP levels in a manner dependent upon *retS* (*Figure 8B–D*, *Figure 8—figure supplement 2*). Moreover, these phenomena were observed at a time point similar to T6S^BT-induced PARA.

Our data suggest that the mechanism underlying PARA is lysis of *P. aeruginosa* cells by a competitor; however, to our knowledge, T4S has not been noted to cause lysis of recipient cells. To determine whether *E. coli* lyse *P. aeruginosa* in a T4S-dependent manner, we quantified extracellular β-galactosidase activity from co-cultures of T4S⁺ and T4S⁻ *E. coli* with *P. aeruginosa attB::lacZ*. Consistent with PARA induction by *E. coli* involving cell lysis, we observed a strong enhancement of *P. aeruginosa* lysis by T4S⁺ *E. coli* (*Figure 8E*). Together, these data demonstrate that the T4SS encoded on the RP4 plasmid induces lysis within a subset of *P. aeruginosa* cells, which in turn induces PARA, leading to H1-T6SS-dependent *E. coli* cell death.

## Discussion

We have shown that self-derived signal(s) generated as a consequence of cell lysis activate the Gac/Rsm pathway of *P. aeruginosa*, and thus stimulate the production of antibiotic factors under its control. In growth competition experiments, the capacity to mount this multifaceted response grants *P. aeruginosa* a significant fitness benefit. Our results suggest that similar to multicellular organisms, injury to a bacterial colony can trigger the release of danger signals that lead to a coordinated response against the threat (*Matzinger, 1994*; *Kaczmarek et al., 2013*). In this study, we focused on offensive factors under control of the Gac/Rsm pathway; however, defensive factors are also likely to be elaborated. For example, a consequence of Gac/Rsm activation in *P. aeruginosa* is the production of c-di-GMP, which activates exopolysaccharide production of *P. aeruginosa* (*Lee et al., 2007*; *Irie et al., 2012*). This increases cellular adhesiveness, which facilitates multicellular aggregates that are more resistant than planktonic cells to an assortment of antibacterial molecules and environmental stresses (*Colvin et al., 2011*; *Billings et al., 2013*). In the context of an infection such as the chronic lung infections that occur in cystic fibrosis patients, host-induced cellular lysis could activate Gac/Rsm and inadvertently convert cells to a state that is more resistant to killing by antibiotics and the immune system.

If PARA was the sole mechanism contributing to enhanced T6S-dependent killing of bacteria with T6 or T4 systems, it would follow that *P. aeruginosa* cells lacking the sensor kinase RetS should target bacteria irrespective of these pathways. However, previous studies have shown that *P. aeruginosa* Δ*retS* retains some ability to differentially target T6S- and T4S-positive vs -negative cells (*LeRoux et al., 2012*; *Basler et al., 2013*). Thus, the response of *P. aeruginosa* to antagonism is comprised of a global response mediated by the Gac/Rsm pathway and a secondary T6S-specific element that is not fully understood. We speculate that this ability of a Δ*retS* strain is related to coordinated spatiotemporal localization of the apparatus among adjacent cells, and that these two mechanisms operate in concert to hone the offensive response of *P. aeruginosa*. PARA may constitute an initial adaptation in which cells perceive a threat in their proximity and increase expression of the H1-T6SS, followed by the orientation of effector translocation specifically toward aggressor cells.

We find that RP4-containing *E. coli* cells induce lysis in *P. aeruginosa*, trigger PARA, and in turn are subject to increased antagonism by the H1-T6SS. This mechanism differs from the model put forth by the Mekalanos laboratory, which suggested that the TPP is required for the response of wild-type *P. aeruginosa* to an incoming conjugative apparatus (*Ho et al., 2013*). A key finding in the prior study was that polymyxin B, an outer membrane-disrupting antibiotic, induces *clpV1* foci formation in wild-type *P. aeruginosa*, but not in a strain lacking *tagT*. This finding, among other data involving strains in the Δ*retS* background of *P. aeruginosa*, led the authors to propose that membrane perturbations caused by an incoming T4 apparatus are sensed by the TPP. A *tagT* deletion strain intrinsically lacks

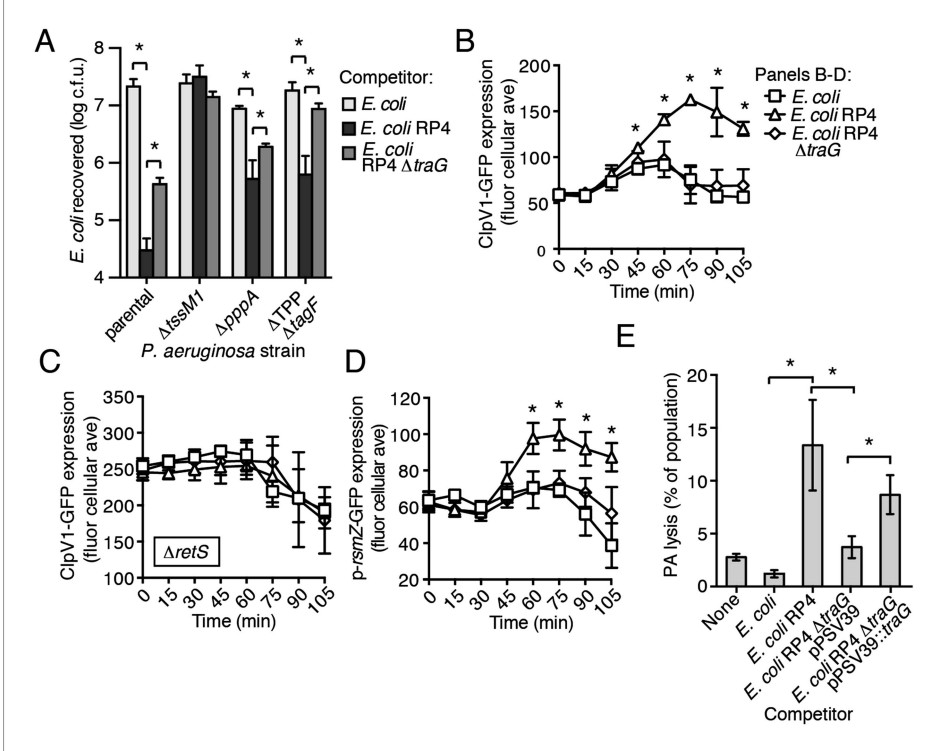

**Figure 8**. The RP4-encoded IncP-type T4SS induces PARA through lysis of *P. aeruginosa*. (**A**) The TPP is not required for differential targeting of *E. coli* with a T4SS. Outcome of growth competition experiments demonstrating increased susceptibility of T4S+ *E. coli* to the H1-T6SS of *P. aeruginosa*. See also *Figure 8—figure supplement 1* for genetic complementation data of the *traG* deletion. n = 4 co-cultures. (**B**) H1-T6SS expression is elevated in the presence of *E. coli* containing a T4SS. Average cellular ClpV1-GFP expression of the *P. aeruginosa clpv1-gfp* throughout co-culture with the indicated *E. coli* strains. (**C**) RetS is required for increased H1-T6SS expression. Average cellular ClpV1-GFP expression of the *P. aeruginosa clpv1-gfp ΔretS* throughout co-culture with the indicated *E. coli* strains. (**D**) *E. coli* bearing a T4SS stimulates *rsmZ* expression. Average cellular GFP levels of *P. aeruginosa* p-*rsmZ-gfp* in TLFM co-culture experiments with the indicated *E. coli* strains. (**B–D**) n = 3 fields; asterisks indicate significant differences between *E. coli* and *E. coli* RP4 co-cultures (p < 0.05). (**E**) The T4SS encoded on RP4 promotes *P. aeruginosa* lysis. Relative *P. aeruginosa attB::lacZ* lysis was measured following co-cultivation with the indicated *E. coli* strain by comparing extracellular to total β-galactosidase activity. Error bars represent ±SD; (**A**) and (**E**) n = 3 co-cultures; asterisks indicate significant differences between indicated groups (p < 0.05).

The following figure supplements are available for figure 8:

**Figure supplement 1**. Genetic complementation of *traG* restores H1-T6SS-dependent targeting of *E. coli* RP4.

**Figure supplement 2**. The RP4-encoded T4SS induces a PARA-associated increase in H1-T6SS activity.

H1-T6SS activity; therefore, interpreting its inability to respond to the antibiotic as evidence of TPP involvement is problematic (*Basler et al., 2013*; *Casabona et al., 2013*; *Ho et al., 2013*). We found that H1-T6SS-active strains that lack the TPP (ΔTPP Δ*tagF*) display a generalized targeting defect, but retain the ability to discriminate T4S+ and T4S− *E. coli*. An alternative explanation for the findings of Ho et al. is that the application of polymyxin B promotes cell lysis, leading to PARA induction (*Barrow and Kwon, 2009*; *Ho et al., 2013*). Attempts to test the validity of this explanation were confounded by pervasive cell death at the antibiotic concentration reported by the authors (20 µg/ml, ~40-fold the minimum inhibitory concentration against *P. aeruginosa* PAO1) (*Barrow and Kwon, 2009*).

The adaptive significance of RP4-induced lysis and its mechanistic basis remain to be resolved. This process could be an altruistic behavior of *P. aeruginosa* that both aborts the T4S-dependent transfer event and alerts surrounding kin, thus decreasing the probability of foreign DNA acquisition by the colony. However, a second possibility is that plasmids such as RP4 carry interbacterial antagonistic

factors that provide fitness to their hosts under certain conditions. It is interesting to note that Ho et al. identified an RP4 transposon mutant that was not targeted by *P. aeruginosa* but retained a functional T4SS (*Ho et al., 2013*). This insertion resides in *trbN*, a gene encoding a periplasmic transglycosylase. It is plausible given the requirement for both T4 structural genes and this peptidoglycan-degrading accessory factor, that the T4S apparatus facilitates the transfer of this protein to recipient cells, where it induces lysis. Alternatively, upon plasmid transfer, the product of *trbN* may stochastically trigger lysis in a small portion of recipient cells.

Efforts to characterize Gac/Rsm-stimulating signal(s) have been performed primarily in *P. fluorescens* (*Lapouge et al., 2008*). This organism is a close relative of *P. aeruginosa*, and the Gac/Rsm pathways of the two species share a number of characteristics including regulation of hydrogen cyanide production and an H1-T6SS-like pathway. Haas et al. have made a number of intriguing observations pertaining to the production and sensing of Gac/Rsm-stimulating signals in *P. fluorescens* that are consistent with our findings in *P. aeruginosa*. Most notably, they found that conditioned media extracts derived from dense cultures of *P. fluorescens*, and, to a lesser extent, *P. aeruginosa* and *Vibrio cholerae*, are sufficient to activate the Gac/Rsm pathway (*Dubuis and Haas, 2007*). In agreement with our results, this indicates that the signal can be self-produced; however, it also raises the intriguing possibility that the lysis of non-self bacteria may activate PARA. While we found that *B. thai*-derived lysate was not sufficient to stimulate PARA, it is possible that other organisms produce the signal. This could lead to a positive feedback loop by which killed competitor cells further stimulate *P. aeruginosa* to produce antibacterial factors.

Despite decades of research, the chemical structure of the molecule(s) that stimulate the Gac/Rsm pathway of *P. aeruginosa*, presumably via interaction with the periplasmic ligand binding domains of its associated sensor kinases, remain unknown. Our own efforts to identify the signaling molecule(s) contained within *P. aeruginosa* lysate, which included assorted enzymatic treatments, have so far been unsuccessful. Structural studies of RetS revealed its periplasmic region bears a fold resembling known carbohydrate interaction domains. A similar domain is predicted in the ecto domain of LadS (*Vincent et al., 2010*). Given that RetS is required for PARA transduction, it is tempting to speculate that cell-associated carbohydrate(s) are released upon lysis and serve as a signal that activates Gac/Rsm in *P. aeruginosa* (*Jing et al., 2010*). The additional observation that extracts derived from multiple bacterial species can stimulate Gac/Rsm in *P. fluorescens* suggests two non-mutually exclusive hypotheses: that the molecule is broadly conserved or that the pathway has evolved to respond to a number of inputs (*Dubuis and Haas, 2007*). The existence of three sensor kinases that operate upstream in the Gac/Rsm pathway of *P. aeruginosa* supports the latter hypothesis.

In what they referred to as 'competition sensing', Cornforth and Foster recently proposed that bacterial stress responses include antagonistic components, and that these pathways have evolved to respond to threats posed by other bacteria (*Cornforth and Foster, 2013*). For example, in response to DNA damage, the SOS pathway stimulates colicin production in *E. coli*, and in *P. aeruginosa*, exogenous peptidoglycan fragments have been shown to stimulate quorum-regulated toxins (*Cascales et al., 2007*; *Korgaonkar et al., 2013*). Our study demonstrates that 'competition sensing' includes an antibacterial response to cellular damage in kin cells. While we found that T6S-dependent killing by *P. aeruginosa* is part of an antagonistic response to lytic threats, Borgeaud et al. reported that in *V. cholerae*, T6S is co-regulated with competence machinery and utilized for obtaining access to exogenous DNA (*Borgeaud et al., 2015*). Together, these studies demonstrate how functionally conserved machinery can be incorporated into diverse cellular programs exhibited by bacteria.

The 'danger theory' of eukaryotic immunity proposes that in addition to the foreign substances that they introduce, threats can be sensed by virtue of cellular damage and ensuing mislocalization of host factors (*Matzinger, 1994*; *Kono and Rock, 2008*). For instance, uric acid microcrystals, which form upon release of the molecule to the sodium-rich extracellular milieu, stimulate dendritic cell maturation (*Shi et al., 2003*). Our study shows that bacteria can also recognize threats by sensing self-derived signals associated with cell damage (*Figure 9*). Moreover, we find that the response to such signals includes the activation of factors that combat the threat—akin to the stimulation of inflammation in eukaryotes. It remains to be determined whether danger sensing is common among bacteria. The Gac/Rsm pathway is conserved widely among Gram-negative γ-proteobacteria; however, the variability of genes under its control confounds a prediction of its general involvement in danger sensing (*Lapouge et al., 2008*). An intriguing possibility is that bacteria can utilize a diversity of signaling systems to sense and respond to kin cell damage.

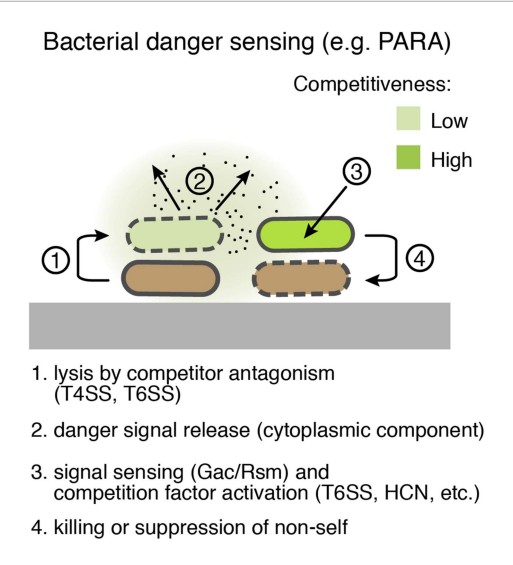

**Figure 9**. Bacterial danger sensing. The model depicts antagonism between two species of bacteria, represented in green and brown. The green cells possess a danger sensing pathway; specifics of PARA are provided in parentheses.

## Materials and methods

### Bacterial strains and culture conditions

All strains used in this study were derived from *P. aeruginosa* PAO1 (*Stover et al., 2000*), *B. thai* E264 (*Yu et al., 2006*), *E. cloacae* ATCC 13047 (*Ren et al., 2010*), and *E. coli* XK1502 (*Panicker and Minkley, 1985*). Routine cultivation of bacteria was performed using Luria broth (LB) medium. A low salt formulation of LB in which no additional sodium chloride was added (LS-LB), was used for plate-based co-culture and competition assays. For *P. aeruginosa*, media were supplemented with 25 µg/ml irgasan, 30 µg/ ml gentamycin, 75 µg/ml tetracycline, or 40 µg/ml X-gal (5-bromo-4-chloro-3-indolyl β-D-galactopyranoside) as necessary, and counter selection for allelic exchange was performed on low-salt LB supplemented with 5% wt/vol sucrose. For *B. thai*, media were supplemented with 25 µg/ml irgasan and 200 µg/ml trimethoprim as necessary, and counter-selection for allelic exchange was performed on M9 minimal medium agar plates containing 0.4% glucose and 0.2% (wt/vol) *p*-chlorophenylalanine (*Chandler et al., 2009*).

For *E. coli*, media was supplemented with 15 µg/ml gentamycin, 200 µg/ml trimethoprim, 25 µg/ml chloramphenicol, and 150 µg/ml carbenicillin as necessary.

### Construction of genetically modified strains

Markerless deletions of genes in *P. aeruginosa* and *B. thai* were generated in frame by allelic exchange with the suicide vectors pEXG2 and pEX18gm (*P. aeruginosa*), or pJRC115 (*B. thai*) (*Rietsch et al., 2005*; *Chandler et al., 2009*). SacB and *pheS*-A304G were used for counterselection in *P. aeruginosa* and *B. thai*, respectively. Deletion alleles were constructed by splicing together 500–600 bp regions flanking the gene to be deleted by overlap extension PCR, and cloning them into the appropriate vector. The open reading frame, except the first and last 4–8 codons of the gene, was replaced by the sequence 5′- TTCAGCATGCTTGCGGCTCGAGTT -3′. A detailed list of strains and vectors used this study are provided in *Tables 1 and 2*, respectively.

Functional translational fluorescent fusions to ClpV1 and Fha1 at their native promoters were achieved by allelic exchange. Constructs were generated by amplification of 500–600 bps regions flanking the C-terminus of the gene. These regions were spliced together with a BamHI site replacing the stop codon and cloned into pEXG2. *Gfp* and *superfolder-gfp (sfgfp)* containing a stop codon was cloned into the BamHI site for *clpV1* and *fha1* constructs, respectively. The translational fusion to MagA was generated using a similar strategy except that only the first 12 codons of *magA* were retained. A resulting clone was introduced to *P. aeruginosa* by allelic exchange. To generate *P. aeruginosa attB*::p-*rsmZ-gfp,* the *rsmZ* promoter was amplified, spliced to an unstable *gfp* variant, and cloned into mini-CTX2 (*Hoang et al., 2000*). The resulting clone was introduced to *P. aeruginosa* by conjugation. The retS[W90A] strain was generated by amplification of 500–600 bps flanking the W90 residue of RetS. The W90A substitution was encoded on the overlap primers and the two regions were spliced together by overlap extension PCR and cloned into pEXG2. The mutation was introduced to PAO1 by allelic exchange. Transcriptional and translational *lacZ* fusions to *tssA1*, original described by Brencic et al. (*Brencic and Lory, 2009*), were introduced to *P. aeruginosa* by conjugation and transformation, respectively. Integration of constitutive fluorescent reporters at the attTn7 site of *B. thai* and *P. aeruginosa* was achieved by four-parental mating or transformation, respectively (*Choi et al., 2005*).

**Table 1.** Strains used in this study

| Organism | Genotype | Reference |
|---|---|---|
| *P. aeruginosa* PAO1 | Type strain | (*Stover et al., 2000*) |
| | attB::*lacZ* | (*Mougous et al., 2006a*) |
| | Δ*tssM1* attB::*lacZ* | This study |
| | Δ*retS* attB::*lacZ* | (*Mougous et al., 2006a*) |
| | Δ*retS* Δ*tssM1* attB::*lacZ* | This study |
| | Δ*tse1* Δ*tsi1* Δ*tse2* Δ*tsi2* Δ*tse3* Δ*tsi3* Δ*tse4* Δ*tsi4* | This study |
| | Δ*retS* Δ*tse1* Δ*tsi1* Δ*tse2* Δ*tsi2* Δ*tse3* Δ*tsi3* Δ*tse4* Δ*tsi4* | This study |
| | *clpV1-gfp* | (*Mougous et al., 2006a*) |
| | *fha1-sfgfp* | This study |
| | *clpV1-gfp* attTn7::*mCherry* | This study |
| | Δ*pppA clpV1-gfp* | (*Silverman et al., 2011*) |
| | ΔTPP Δ*tagF clpV1-gfp* | This study |
| | Δ*pppA* | (*Mougous et al., 2007*) |
| | ΔTPP Δ*tagF* | (*Silverman et al., 2011*) |
| | Δ*tssM1* | (*Silverman et al., 2011*) |
| | attTn7::*PA0082-lacZ10-Gm* | (*Brencic and Lory, 2009*) |
| | attB::*PA0082-lacZ-tet* | (*Brencic and Lory, 2009*) |
| | p-*rsmZ-gfp* attTn7::*Gm-mCherry* | This study |
| | Δ*retS clpV1-gfp* | (*LeRoux et al., 2012*) |
| | Δ*ladS clpV1-gfp* | This study |
| | Δ*gacS clpV1-gfp* attTn7::*Gm-mCherry* | This study |
| | p-*magA-sfgfp* | This study |
| | *retS*$^{W90A}$ | This study |
| | Δ*gacS* | This study |
| | attTn7::*Gm-gfp* | (*LeRoux et al., 2012*) |
| | attTn7:: *Gm-mCherry* | (*LeRoux et al., 2012*) |
| | Δ*tssM1* attTn7::*Gm-mCherry* | This study |
| | Δ*gacS* attTn7:: *Gm-mCherry* | This study |
| *B. thailandensis* E264 | Type strain | (*Yu et al., 2006*) |
| | Δ*tssM-1* | This study |
| | attTn7::*Tp-P$_{S12}$-mCherry* | (*LeRoux et al., 2012*) |
| | Δ*tssM-1* attTn7::*Tp-P$_{S12}$-mCherry* | This study |
| | attTn7::*Tp-P$_{S12}$-GFP* | (*LeRoux et al., 2012*) |
| | Δ*tssM-1* attTn7::*Tp-P$_{S12}$-GFP* | This study |
| | attTn7::*Tp-CFP* | This study |
| *E. cloacae* ATCC 13047 | Type strain | (*Ren et al., 2010*) |
| | Δ*tssM* | (*Whitney et al., 2014*) |
| *E. coli* XK1502 | Type strain | (*Panicker and Minkley, 1985*) |
| | RP4 | (*Pansegrau et al., 1994*) |
| | RP4 Δ*traG* | This study |
| | RP4 Δ*traG* pPSV39 | This study |
| | RP4 Δ*traG* pPSV39::*traG* | This study |

**Table 2**. Plasmids used in this study

| Plasmid | Utility | Reference |
|---|---|---|
| **P. aeruginosa PAO1** | | |
| miniCtx::lacZ | constitutive LacZ expression | (*Vance et al., 2005*) |
| pUC18T-miniTn7T-Gm-gfp | constitutive GFP expression | (*Choi et al., 2005*) |
| pUC18T-miniTn7T-Gm-mCherry | constitutive mCherry expression | (*LeRoux et al., 2012*) |
| pEXG2_ΔPA4856 | *retS* deletion allele | (*Mougous et al., 2006a*) |
| pEXG2_ΔPA0077 | *tssM1* deletion allele | (*Mougous et al., 2006a*) |
| pEXG2_ΔPA1844-5 | *tse1 tsi1* deletion allele | (*Russell et al., 2011*) |
| pEXG2_ΔPA2702-3 | *tse2 tsi2* deletion allele | (*Hood et al., 2010*) |
| pEXG2_ΔPA3484-5 | *tse3 tsi3* deletion allele | (*Russell et al., 2011*) |
| pEXG2_ΔPA2774-5 | *tse4 tsi4* deletion allele | (*Whitney et al., 2014*) |
| pEXG2_PA0090-gfp | *clpV1* functional translational GFP fusion allele | (*Mougous et al., 2006a*) |
| pEXG2_PA0081-sfgfp | *fha1* functional translational GFP fusion allele | This study |
| pEXG2_ΔPA0075 | *pppA* deletion allele | (*Mougous et al., 2007*) |
| pEXG2_ΔPA0070-0076 | *tagQRST ppkA pppA tagF* deletion allele | (*Silverman et al., 2011*) |
| attTn7::PA0082-lacZ10-Gm | *tssA1* translational *lacZ* reporter | (*Brencic and Lory, 2009*) |
| attB::PA0082-lacZ-tet | *tssA1* transcriptional *lacZ* reporter | (*Brencic and Lory, 2009*) |
| miniCtx_p-PA3621.1-gfp | *rsmZ* transcriptional GFP reporter | This study |
| pEXG2_ΔPA0928 | *gacS* deletion allele | This study |
| pEXG2_ΔPA3974 | *ladS* deletion allele | This study |
| pEXG2_p-PA4492-gfp | *magA* translational GFP fusion allele | This study |
| pEXG2_PA4856$^{W90A}$ | *retS$^{W90A}$* allele | This study |
| **B. thailandensis E264** | | |
| pJRC115_ΔBTH_I2954 | *tssM-1* deletion allele | This study |
| pUC18T-miniTn7T-Tp-PS12-gfp | constitutive GFP expression | (*Schwarz et al., 2010*) |
| pUC18T-miniTn7T-Tp-PS12-mCherry | constitutive mCherry expression | (*LeRoux et al., 2012*) |
| pUC18T-miniTn7T-Tp-ecfp | Constitutive CFP expression | (*Choi et al., 2005*) |
| **E. coli XK1502** | | |
| RP4 | Naturally occurring plasmid encoding IncP-type T4SS | (*Pansegrau et al., 1994*) |
| E. coli RP4 Δ*traG* | RP4 bearing *traG* deletion | This study |
| pPSV39 | Expression vector | (*Silverman et al., 2013*) |
| pPSV39-*traG* | IPTG-inducible TraG expression for complementation | This study |

*TraG* on RP4 was replaced with a chloramphenicol resistance cassette by λ Red recombination (*Datsenko and Wanner, 2000*) in *E. coli* CC1254, resulting in RP4 Δ*traG*::chlmR (RP4 Δ*traG*). Following PCR confirmation, RP4 Δ*traG* was transferred to XK1502 by P1 phage transduction.

## Time-lapse fluorescence microscopy

Overnight cultures were diluted 1:50 or 1:100 in LB and incubated with aeration at 37°C until an $OD_{600}$ of 0.5–0.7 was reached. Cultures were concentrated fivefold and mixed 1:1 by volume with the indicated competitor. 1–2 µl of the bacterial suspension was spotted onto a 1.5% wt/vol agarose

growth pads (prepared using Vogel Bonner minimal media containing 0.2% wt/vol sodium nitrate and 0.01% wt/vol casamino acids) and sealed.

Microscopy data were acquired using NIS Elements (Nikon) acquisition software on a Nikon Ti-E inverted microscope with a 60× oil objective, automated focusing (Perfect Focus System, Nikon), a xenon light source (Sutter Instruments), and a CCD camera (Clara series, Andor). Lysis and growth rate were measured from TLFM sequences acquired at 5-min intervals; expression was measured from 15-min interval TLFM sequences. At least three fields were acquired and analyzed for each experimental group, and experiments were performed independently multiple times.

## Image analysis

TLFM sequences were analyzed using previously described methods (*LeRoux et al., 2012*). Briefly, cells were identified from phase images using a watershed algorithm. *P. aeruginosa* and competitor cell populations were distinguished by the constitutive expression of cytoplasmic mCherry in either *P. aeruginosa* or the competitor organism. Non-cell debris was excluded based on a size threshold and/or fluorescence gating. For display in figures only, the GFP channel was γ-transformed (ClpV1-GFP, exponent = 1.2; Fha1-GFP, exponent = 1.3), thresholded, and the Matlab colormap (jet) was applied.

To calculate average cellular fluorescence, GFP intensity of non-cell regions was subtracted from GFP intensity of cell regions as defined by cell masks. To identify foci within cells, the cytoplasmic fluorescence of each cell was fit to an empirical model for cytoplasmic fluorescence. A cell template image was generated by applying a Gaussian blur (radius 2 pixels) to the square root of the distance transform applied to the cell mask. A least-squares fit of a constant times the cell template was then performed to the observed cell image. The noise was computed as the standard deviation of the cell intensity after subtracting the fits of the cytoplasmic fluorescence and the fits of all potential foci. The signal to noise of a potential focus was computed as the intensity of the brightest pixel divided by the noise of that cell, and was defined as a focus if the value exceeded an empirically determined threshold. Candidate foci were identified globally using a watershed algorithm, then assigned to cells whose masks overlapped the focus position. The average and standard deviation of the percentage of *P. aeruginosa* cells with foci calculated for three fields is plotted.

For contact-dependence analyses, cell neighbors were defined as cells with a boundary within a 2-pixel radius of the current cell (*LeRoux et al., 2012*). Growth rate and cellular lysis were determined from datasets in which cells were linked over time based on frame-to-frame overlap of bright-field images. Doubling times—defined as minutes between birth and death—were calculated for cells that arose from a division after the start of the experiment and divided prior to the end of the experiment. A lysis event was defined as an 80% decrease in fluorescence intensity of a single cell between consecutive frames and was measured for in strains expressing a constitutive fluorophore (mCherry, GFP, or CFP). For figures in which end-point *P. aeruginosa* lysis was presented, the number of *P. aeruginosa* cells that lysed was normalized to the initial number of *P. aeruginosa*–*B. thai* contacts to control for small fluctuations in cell density and *P. aeruginosa*–competitor ratio.

## Growth competition assays

For all end-point growth competition assays, except when noted, overnight cultures were used. Cultures were washed and mixed at the indicated volumetric ratios. Five microliters of the resulting mixture was spotted on a nitrocellulose membrane placed on a LS-LB 3% wt/vol agar plate and incubated at 37°C. For intraspecies growth competition experiments, *P. aeruginosa attB::lacZ* was used as the donor strain background, experiments were initiated at a 1:1 donor: recipient ratio, and were plated on LB containing X-gal for enumeration. Growth competition experiments with *B. thai* competitors were initiated at a donor:recipient ratio of 5:1 (interspecies wild-type and T6S-dependent fitness, *Figure 1*; TPP strain experiments, *Figure 3*) or 1:1 (Δ*tssM1* and Δ*gacS* fitness, *Figure 5A*) and plated on LB containing gentamycin (30 µg/ml) for *B. thai* selection and trimethoprim (200 µg/ml) for *P. aeruginosa* selection. For *E. cloacae* competitors, which encode *lacZ*, log-phase cultures were mixed at a ratio of 8:1 (TPP strain experiments, *Figure 3—figure supplement 2*) or 1:1 (Δ*tssM-1* and Δ*gacS* fitness, *Figure 5B*) and plated on media containing X-gal for *P. aeruginosa*–*E. cloacae* enumeration. For growth competition experiments with *E. coli* competitors, a *P. aeruginosa attB::lacZ* background was used,

experiments were initiated at a donor:recipient ratio of 1:2, incubated for 3 hr, and plated on LB containing X-gal.

## H1-T6SS transcriptional and translational assays

*P. aeruginosa* strains bearing previously validated chromosomally encoded transcriptional or translation fusions to *tssA1* (PA0082) were utilized to quantify transcription and translation (see also *Tables 1 and 2*) (*Brencic and Lory, 2009*). Overnight cultures were washed, mixed at a 1:2 ratio with media (monoculture), *B. thai*, or *B. thai ΔtssM-1,* then spotted on a nitrocellulose membrane placed on a LS-LB 3% wt/vol agar plate. Following a 3 hr incubation at 37°C, cells were harvested in PBS, washed, and assayed for relative levels of β-galactosidase activity using the Galacto-Light Plus Reporter Gene Assay System (Life Technologies).

## Diffusion experiments

Bacterial suspensions composed of overnight cultures, washed once and mixed at a 2:1 *P. aeruginosa*:*B. thai* ratio were spotted as depicted in *Figure 6C* on LS-LB 3% wt/vol agar. Following 3 hr of incubation at 37°C, bacteria growing on top of the filter (*Figure 6C*, black dashed lines) were resuspended in LB and imaged by microscopy to determine PARA induction. To exclude dead *P. aeruginosa* cells from our analysis, which were prevalent in Condition 1, a *P. aeruginosa* strain constitutively expressing mCherry was used and only mCherry-positive cells were considered.

## Lysate experiments

Stationary phase *P. aeruginosa* or *B. thai* cultures were pelleted and resuspended in growth media or PBS before sonication. Colony forming units (c.f.u.) of cultures and lysates were enumerated to verify that >95% of cells were lysed. For TLFM experiments, an agarose pad prepared as described above was infused with either lysate or PBS (control). The concentration of lysate in the agarose pad was approximately $5 \times 10^4$ lysed cells/μl. Reporter strains were cultivated to $OD_{600}$ of 0.5–0.7, spotted on the lysate-containing agarose pad, and imaged. For the growth competition assays, lysate or PBS (control) was spotted on a LS-LB 3% wt/vol agar growth plate, a nitrocellulose filter was placed on top, and indicated mixtures of cells were spotted directly over the lysate.

## Sample preparation for proteomic analysis

LB plates were prepared by applying either lysate (25-fold concentrated *P. aeruginosa* at an $OD_{600}$ of 0.5–0.7 resuspended in PBS and lysed by sonication) or PBS (control) at 2% (vol/vol). A concentrated *P. aeruginosa clpV1-gfp* culture (at $OD_{600}$ 0.5–0.7) was spotted on the nitrocellulose membrane placed on each plate then incubated for 2.5 hr at 37°C. Cells were harvested in PBS, washed, then stored at −80°C. Duplicate biological samples were prepared for mass spectrometry analysis as described previously (*Whitney et al., 2014*).

The semi-quantitative technique of spectral counting was used to determine the relative abundance of identified proteins in each sample as described in Whitney et al. (*Liu et al., 2004*; *Whitney et al., 2014*). Only proteins present in both biological replicates and with a sum of 20 spectral counts or greater across all replicates were considered in our analysis.

## T4S-dependent lysis of *P. aeruginosa*

Overnight cultures of *P. aeruginosa attB::lacZ* were washed, mixed 1:2 with *E. coli* strains, and spotted on a nitrocellulose membrane placed on a NS-LB 3% wt/vol agar plate. Following 3 hr of incubation at 37°C, levels of extracellular and total β-galactosidase activity were determined as previously described (*Chou et al., 2012*). The percentage of lysed *P. aeruginosa* lysis was determined by normalizing extracellular to total β-galactosidase activity.

## Statistical tests

All TLFM datasets were analyzed by two-way ANOVA using a Bonferroni correction for testing multiple hypotheses. End-point assays were analyzed using a two-tailed Student's *t* test. Asterisks indicate significance at $p < 0.05$. Proteomic data were analyzed using Gene Set

Enrichment Analysis (GSEA) (*Mootha et al., 2003*; *Subramanian et al., 2005*). Genes positively regulated by the Gac/Rsm pathway were defined as those found differentially regulated in a prior study comparing mRNA levels of Δ*retS* and wild-type *P. aeruginosa* (*Goodman et al., 2004*). 1000 permutations of the analysis were performe d across both phenotype and gene set, and the ratio of classes metric was used for ranking genes.

## Acknowledgements

We thank D Low and C Hayes for *E. cloacae* strains, J Harrison for the *gacS* deletion construct, S Lory for β-galactosidase reporters, A Rietsch and P de Boer for codon-optimized superfolder *gfp*, H Schweizer for Tn7 constructs, and N Kuwada and H Kulasekara for helpful discussions. ML was supported by the Molecular and Cellular Biology Training Grant (T32GM007270) and EIM was a University of Washington Mary Gates Scholar. JDM holds an Investigator in the Pathogenesis of Infectious Disease Award from the Burroughs Wellcome Fund.

## Additional information

### Funding

| Funder | Grant reference number | Author |
| --- | --- | --- |
| National Institutes of Health (NIH) | AI105268 | Paul A Wiggins, Joseph D Mougous |
| Cystic Fibrosis Foundation (CFF) | CFR565-CR07 | Joseph D Mougous |
| National Institutes of Health (NIH) | Cellular and Molecular Training Grant, T32GM007270 | Michele LeRoux |
| National Institutes of Health (NIH) | AI080609 | Joseph D Mougous |
| University of Maryland, Baltimore County | SOP1841-IQB2014 School of Pharmacy Mass Spectrometry Center | David R Goodlett |

The funders had no role in study design, data collection and interpretation, or the decision to submit the work for publication.

### Author contributions

MLR, JDM, Conception and design, Acquisition of data, Analysis and interpretation of data, Drafting or revising the article; RLK, Conception and design, Acquisition of data, Analysis and interpretation of data; EIM, BQT, BNH, JCW, ABR, YAG, DRG, Acquisition of data, Analysis and interpretation of data; SBP, Acquisition of data, Analysis and interpretation of data, Drafting or revising the article; BT, Acquisition of data; PAW, Conception and design, Analysis and interpretation of data

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
