## [Decision Letter]

Thank you for sending your work entitled “Kin cell lysis is a danger signal that activates antibacterial pathways of *Pseudomonas aeruginosa*” for consideration at *eLife*. Your article has been favorably evaluated by Michael Marletta (Senior editor) and 3 reviewers, one of whom, Michael Laub, is a member of our Board of Reviewing Editors.

The Reviewing editor and the other reviewers discussed their comments before we reached this decision, and the Reviewing editor has assembled the following comments to help you prepare a revised submission.

Each of the three reviewers thought this was a well-done study that uncovered an exciting and interesting new aspect of T6SS biology. The reviewers were generally satisfied with the data and the conclusions drawn from these data, but each raised a number of points that should be addressed in a revised manuscript. Most are issues about the clarity of presentation, but there are a couple of cases in which additional experiments are likely required.

Reviewer #1

Figure 4: The description of the role of RetS was confusing, as the text says retS is required for PARA. Initially I thought that implied RetS was required for induction of the genes involved in PARA. But actually, in the *retS* mutant the genes for T6SS and PARA are constitutively produced (Figure 4). So while *retS* is required for “signal-responsiveness”, the way it's written confuses the reader into thinking *retS* is required for transcription of *clpV1-gfp* and other readouts that report on the transcription of the T6SS apparatus.

Further, in all likelihood, the signal sensed by RetS likely antagonizes it from antagonizing GacS, thereby activating GacA. It might help the reader if the authors just spelled out this idea/working model more explicitly and in plain language earlier on.

Figure 7: I don't understand the units in 7A which indicates 'lysis events'. The text says it's a small percentage but the figure isn't actually reporting percentages. This needs clarification.

Figure 7: Can lysis events be visualized or seen in the time-lapse movies shown earlier? This seems essential to show.

Although 52% of Gac targets also are induced by lysate (Figure 7), that means nearly 50% did not respond—why not? This should be commented on.

The authors pose a reasonable solution to the discrepancy between their model and that previously proposed, namely that polymyxin B may have lead previously to cell lysis and induction of PARA. It would seem easy enough and well worth testing (if only to cite as data not shown) whether this is the case.

Reviewer #2

1) The authors present strong evidence that PARA is mediated through Gac (Figure 4). However, I think it's important they explicitly demonstrate that PARA does not occur in a GacA or GacS mutant. This could be done by analyzing expression of the rsmZ-GFP reporter in response to *B. thai* (as in Figure 4) in a *gacA* or *gacS* mutant background.

2) Induction of PARA appears to occur in only a subpopulation of cells (Figure 2; Figure 4). The authors should comment on this heterogeneity; plotting the expression of ClpV1-gfp (for e.g.) on the basis of average fluorescent cell intensity (as in Figure 2), may obscure the difference in expression between cells in which the system is induced and those in which the system is not. The degree of induction of the system at the level of the individual cell, as opposed to the population as a whole, is therefore likely to be much higher than reported.

3) The authors should consider testing the prediction that lysis of kin cells occurs/begins prior to PARA if technically feasible. Currently they show that *B. thai* results in kin cell lysis, that kin cell lysis induces PARA, and that lysis of kin cells precedes lysis of *B. thai* competitor cells.

Reviewer #3

Overall the experiments described are rigorous and well-designed, and the conclusions are supported by the data shown. My only criticism is that many of the experiments lack sufficient descriptive detail (either in the Methods, Results or figure legends) to fully understand what was done. For example:

1) The “T6S-dependent fitness” displayed by *P. aeruginosa* (in Figure 1 and in the first paragraph of the Results section) should be better defined in the Results section and the way the experiments was conducted should be better described in the Materials and methods section.

2) Figure 2: It is unclear whether the fluorescent reporters shown in the TLFM images (Figure 2) are the same as the expression reporters quantified in the accompanying graphs (Figure 2), since they are labeled slightly differently (ClpV1-GFP vs *clpV1-gfp*). Similarly, the Methods section describing the construction of these reporters needs more detail. Although they were constructed according to previously published methods, a brief description should be included here. Also no reference is cited for the statement that these were generated as previously described.

3) A description of the transcriptional and translational *lacZ* reporters shown in Figure 4 should be included in the Methods. Also, it is unclear how the “LacZ” activity measured from these mixed populations was normalized and what *P. aeruginosa* strain monoculture was used for normalization.

---

## [Author Response]

Reviewer #1

Figure 4*: The description of the role of RetS was confusing, as the text says retS is required for PARA. Initially I thought that implied RetS was required for induction of the genes involved in PARA. But actually, in the* retS *mutant the genes for T6SS and PARA are constitutively produced (*Figure 4*). So while* retS *is required for* “*signal-responsiveness*”*, the way it's written confuses the reader into thinking* retS *is required for transcription of clpV1-gfp and other readouts that report on the transcription of the T6SS apparatus.*

*Further, in all likelihood, the signal sensed by RetS likely antagonizes it from antagonizing GacS, thereby activating GacA. It might help the reader if the authors just spelled out this idea/working model more explicitly and in plain language earlier on*.

We thank the reviewer and appreciate that this is a potentially confusing point. We have changed our wording in this section to clarify the proposed function of RetS (paragraph 8 of the Results section).

Figure 7*: I don't understand the units in 7A which indicates 'lysis events'. The text says it's a small percentage but the figure isn't actually reporting percentages. This needs clarification*.

We have corrected the text to reflect that we are measuring the number of *P. aeruginosa* cells that lyse (paragraph 11 in the Results section). Additionally, we have normalized lysis to reflect the initial cell density of the experiment.

Figure 7*: Can lysis events be visualized or seen in the time-lapse movies shown earlier? This seems essential to show*.

We thank the reviewer for this suggestion, which was also made by reviewer 2. To address this point, we concurrently measured *P. aeruginosa* lysis, *B. thai* lysis, and ClpV1-GFP levels in single TLFM series. Consistent with our model, we found that *P. aeruginosa* lysis commences shortly after the experiment is initiated, prior to induction of ClpV1-GFP expression (Figure 7, Video 6). We also observed that the increase in ClpV1-GFP is coincident with increased *B. thai* lysis.

*Although 52% of Gac targets also are induced by lysate (*Figure 7*), that means nearly 50% did not respond*—*why not? This should be commented on*.

There appears to be some confusion in this instance. We stated that ∼50% of proteins overrepresented in abundance in the lysate treated sample are known Gac/Rsm targets, not that 50% of all Gac/Rsm targets were overrepresented. Our proteome experiment identified roughly 20% of the hypothetical *P. aeruginosa* proteome (1139 proteins of the 5639). There are 257 proteins negatively regulated by Gac/Rsm, which constitutes 4.5% of the proteome. Of the 1139 proteins we identified, 4.5% were Gac targets, demonstrating that while our sampling was not exhaustive, it was random and representative of the proteome as a whole. Gene set enrichment statistical analysis yields credence to our hypothesis that the Gac/Rsm pathway is stimulated in the presence of *P. aeruginosa* lysate, which was what this global analysis of the proteome was intended to test.

*The authors pose a reasonable solution to the discrepancy between their model and that previously proposed, namely that polymyxin B may have lead previously to cell lysis and induction of PARA. It would seem easy enough and well worth testing (if only to cite as data not shown) whether this is the case*.

We agree with the reviewer and thus attempted to repeat the study conducted by Ho et al. However, cell death began immediately following the addition of 20 µg/mL polymyxin B (as described in Ho et al.) and >90% of cells died prior to the time needed for PARA induction (∼15-30 min). This result casts doubt on the involvement of PARA in the process described by Ho et al.Author response image 1.*P. aeruginosa mCherry* cells 30 minutes after exposure to antibiotic. Cells lose fluorescence upon permeabilization and death.

Reviewer #2

*1) The authors present strong evidence that PARA is mediated through Gac (*Figure 4*). However, I think it's important they explicitly demonstrate that PARA does not occur in a GacA or GacS mutant. This could be done by analyzing expression of the rsmZ-GFP reporter in response to B. thai (as in*
Figure 4*) in a* gacA *or* gacS *mutant background.*

We agree with the reviewer that this is an important experiment. To be consistent with analogous experiments involving *retS* and *ladS*, we generated a ∆*gacS clpV1-gfp* reporter strain and found that, as predicted, a strain lacking *gacS* is unable to respond to the presence of *B. thai* with an active T6SS (Figure 4).

*2) Induction of PARA appears to occur in only a subpopulation of cells (*Figure 2*;*
Figure 4*). The authors should comment on this heterogeneity; plotting the expression of ClpV1-gfp (for e.g.) on the basis of average fluorescent cell intensity (as in*
Figure 2*), may obscure the difference in expression between cells in which the system is induced and those in which the system is not. The degree of induction of the system at the level of the individual cell, as opposed to the population as a whole, is therefore likely to be much higher than reported*.

To assess the heterogeneity of PARA within a population of *P. aeruginosa*, we generated a histogram of cellular ClpV1-GFP expression of *P. aeruginosa–B. thai* co-culture from a time point when distinct changes are detected (90 minutes). We find that the distribution remains normal, but that the mean is shifted (Figure 2—figure supplement 2). This distribution is consistent with a graded increase in expression, as opposed to a binary switch.

*3) The authors should consider testing the prediction that lysis of kin cells occurs/begins prior to PARA if technically feasible. Currently they show that B. thai results in kin cell lysis, that kin cell lysis induces PARA, and that lysis of kin cells precedes lysis of B. thai competitor cells*.

The proposed experiment, which was also suggested by reviewer 1, is an excellent suggestion and has been added as Figure 7 and Video 6.

Reviewer #3

*Overall the experiments described are rigorous and well-designed, and the conclusions are supported by the data shown. My only criticism is that many of the experiments lack sufficient descriptive detail (either in the Methods, Results or figure legends) to fully understand what was done*.

We have reorganized the Materials and methods section to better clarify each experiment and have added additional details regarding the experiments and strains used.

*For example*:

*1) The* “*T6S-dependent fitness*” *displayed by* P. aeruginosa *(in*
Figure 1
*and in the first paragraph of the Results section) should be better defined in the Results section and the way the experiments was conducted should be better described in the Materials and methods section.*

We apologize for the confusion and have improved our description of this label in the revised figure legend. Also, at the suggestion of the reviewer, we have added a description of the experiment to the Materials and methods section of the manuscript .

*2)*
Figure 2*: It is unclear whether the fluorescent reporters shown in the TLFM images (*Figure 2*) are the same as the expression reporters quantified in the accompanying graphs (*Figure 2*), since they are labeled slightly differently (ClpV1-GFP vs* clpV1-gfp*). Similarly, the Methods section describing the construction of these reporters needs more detail. Although they were constructed according to previously published methods, a brief description should be included here. Also no reference is cited for the statement that these were generated as previously described.*

The reviewer is correct that the graphs in Figure 2 present quantification of the datasets summarized in Figure 2. We have improved the consistency of our reporter nomenclature to avoid this confusion. Descriptions of strain construction and the omitted references have been added to the Materials and methods section and to the new table summarizing plasmids used in this study (Table 2), respectively.

*3) A description of the transcriptional and translational* lacZ *reporters shown in*
Figure 4
*should be included in the Methods. Also, it is unclear how the* “*LacZ*” *activity measured from these mixed populations was normalized and what* P. aeruginosa *strain monoculture was used for normalization.*

The transcriptional and translational H1-T6SS *lacZ* reporters were referenced in the strains table of the original submission. At the request of the reviewer, we now also cite the paper describing their construction in the Materials and methods section. We have also clarified and expanded upon our original description of our assays to measure H1-T6SS expression using these reporters in a new Materials and methods subsection titled “H1-T6SS transcriptional and translational assays”.